# Chromatin accessibility during human first-trimester neurodevelopment

Camiel C. A. Mannens[1], Lijuan Hu[1], Peter Lönnerberg[1], Marijn Schipper[2], Caleb C. Reagor[3], Xiaofei Li[4], Xiaoling He[5], Roger A. Barker[5], Erik Sundström[4], Danielle Posthuma[2] & Sten Linnarsson[1✉]

The human brain develops through a tightly organized cascade of patterning events, induced by transcription factor expression and changes in chromatin accessibility. Although gene expression across the developing brain has been described at single-cell resolution[1], similar atlases of chromatin accessibility have been primarily focused on the forebrain[2–4]. Here we describe chromatin accessibility and paired gene expression across the entire developing human brain during the first trimester (6–13 weeks after conception). We defined 135 clusters and used multiomic measurements to link candidate *cis*-regulatory elements to gene expression. The number of accessible regions increased both with age and along neuronal differentiation. Using a convolutional neural network, we identified putative functional transcription factor-binding sites in enhancers characterizing neuronal subtypes. We applied this model to *cis*-regulatory elements linked to *ESRRB* to elucidate its activation mechanism in the Purkinje cell lineage. Finally, by linking disease-associated single nucleotide polymorphisms to *cis*-regulatory elements, we validated putative pathogenic mechanisms in several diseases and identified midbrain-derived GABAergic neurons as being the most vulnerable to major depressive disorder-related mutations. Our findings provide a more detailed view of key gene regulatory mechanisms underlying the emergence of brain cell types during the first trimester and a comprehensive reference for future studies related to human neurodevelopment.

Through a tightly organized cascade of patterning, specification and differentiation events, the human brain develops into a highly complex system capable of unique cognitive abilities beyond those of other mammals. The human brain consists of more than 1,000 distinct types of neuron, glia and non-neural cell types[5]. Single-cell RNA sequencing (scRNA-seq) has enabled parallel profiling of cell types and states, revealing both regional differences and subtle variation between closely related cell types[6–8]. Profiling the developing human brain has revealed differentiation trajectories leading to diverse neuronal and non-neuronal cell types[1]. During development, the functional architecture of the genome is constantly in flux, with changes in the expression, binding and regulation of transcription factors (TFs) driving cell-fate decisions. The activities of regulatory elements in development are often both cell-type specific and brief. This dynamism complicates the interpretation of genome-wide association studies (GWAS) of complex neurodevelopmental disorders, because identified loci—which predominantly fall in the non-coding DNA—are equally context specific[9,10]. Previous work has mapped the regulatory landscapes of the developing human brain in the second-trimester developing cortex[2,3], whole embryos[4], as well as organoids and induced pluripotent stem cell-derived model systems[11]. Here we focus on the chromatin landscape across the whole developing human brain during the first trimester, a pivotal time when the brain is patterned and many neural cell types acquire their core transcriptional identities.

## Chromatin accessibility in the first trimester

We measured chromatin accessibility in the developing human brain from 6 to 13 post-conception weeks using the 10x Genomics single-cell assay of transposase-accessible chromatin using sequencing (scATAC-seq[12]; 18 specimens), a combined scATAC-seq and scRNA-seq assay (multiome; 3 specimens) or both (5 specimens; Fig. 1a,b). Each specimen was dissected into the main antero-posterior segments (telencephalon, diencephalon, mesencephalon, metencephalon and cerebellum; Fig. 1c). We collected relatively more nuclei from the brain stem region, which is highly complex but has been comparatively less studied than the forebrain[2–4]. After removing low-quality nuclei (Methods), we collected chromatin profiles from a total of 526,094 nuclei and 76 unique biological samples (116 including technical replicates; Extended Data Fig. 1 and Supplementary Table 1). A total of 166,785 of these nuclei included gene expression profiles from multiome sequencing.

[1]Division of Molecular Neurobiology, Department of Medical Biochemistry and Biophysics, Karolinska Institute, Solna, Sweden. [2]Department of Complex Trait Genetics, Center for Neurogenomics and Cognitive Research (CNCR), Vrije Universiteit Amsterdam, Amsterdam, The Netherlands. [3]Howard Hughes Medical Institute and Laboratory of Sensory Neuroscience, The Rockefeller University, New York, NY, USA. [4]Division of Neurodegeneration, Department of Neurobiology, Care Sciences and Society, Karolinska Institutet, Solna, Sweden. [5]John van Geest Centre for Brain Repair, Department of Clinical Neurosciences, Wellcome-MRC Cambridge Stem Cell Institute, University of Cambridge, Cambridge, UK. ✉e-mail: sten.linnarsson@ki.se

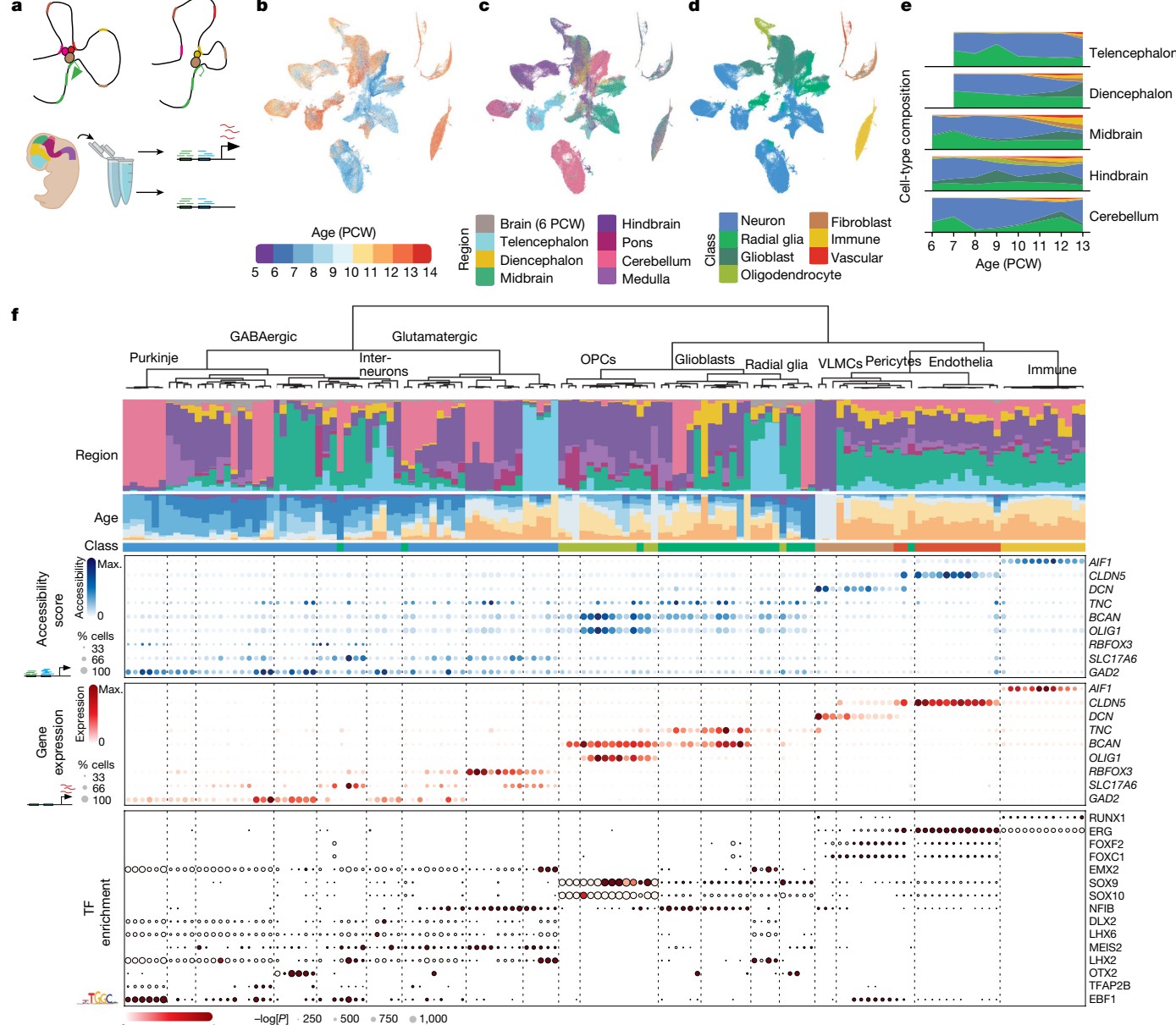

**Fig. 1 | Atlas overview. a**, Overview of experimental design. Dissected samples were processed using scATAC-seq and single-cell multiome sequencing to infer gene-regulatory relationships throughout early development across the human brain. Illustration by Ivana Kapustova. **b–d**, *t*-SNE embedding plot of developmental ages (**b**), regional identities (**c**) and cell classes (**d**). PCW, post-conception week. **e**, Proportions of cell class by age and anatomical region. **f**, Top to bottom: per-cluster regional distribution of cell types (legend in **c**); distribution of ages (legend in **b**); assigned cell class (legend in **d**); aggregated gene activity by cluster based on region co-accessibility; gene expression of the same genes; TF motif enrichment (HOMER; one-sided; no multiple test correction) in the top 2,000 most enriched accessible regions per cluster (by Pearson residual enrichment). For the accessibility scores and gene expression, the dot size represents percentage of positive cells. For motif enrichment, the dot size represents the *P* value and colour indicates corresponding expression of the TF (trinarization score; a probabilistic score of whether a gene is expressed). OPCs, oligodendrocyte progenitor cells; VLMCs, vascular lepotomeningeal cells; max., maximum.

To identify the feature set of accessible regions, we applied stratified peak calling on a rough clustering of the data using 20-kb genomic bins as temporary features. This was followed by a more robust clustering based on the accessible regions using latent semantic indexing and Louvain clustering. Batch correction was carried out using Harmony. A split-and-pool approach was then used to subcluster each cell class (radial glia or glioblast, oligodendrocyte progenitor cell, neuron, fibroblast, vascular and immune cells; Fig. 1d,e, Extended Data Fig. 2 and Supplementary Table 2), resulting in 135 clusters (Fig. 1f).

Following annotation of the accessible regions (Extended Data Fig. 3), about 36% of accessible regions were found to be intergenic,

whereas 64% were in gene bodies or promoter regions with most being intronic (51%; Extended Data Fig. 3a). When taking only the distance to the transcription start site (TSS) into account, 87% of accessible regions were marked as distal (>2 kb from the TSS) and 19,494 accessible regions (4.7%) overlapped with known TSS sites (Extended Data Fig. 3b). Additionally, 18% of elements overlapped with a transposable element, compared to 22% in similar data from the adult human brain[13] (Extended Data Fig. 3i).

Access to gene expression data also allowed for better identification of candidate *cis*-regulatory elements (cCREs) using a modified version of Cicero. By leveraging co-occurrence of chromatin accessibility and

gene expression, 106,991 predicted enhancer–gene interactions were identified for 16,267 genes and 59,069 accessible regions (henceforth, cCREs).

The dataset primarily consisted of nuclei from the neural lineage, showing strong regional identities, whereas non-neural clusters including endothelial cells, fibroblasts and microglia showed limited spatial identities (Extended Data Fig. 4a,b). Radial glia are the progenitor cells that give rise to neurons, before transitioning to a more restricted glioblast identity that gives rise to oligodendrocyte progenitor cells and astrocytes. The ratio of radial glia to glioblasts differed markedly between regions, with posterior regions being enriched for glioblasts whereas the more anterior regions showed primarily radial glia (Extended Data Fig. 4c; see Extended Data Fig. 4d–f for further information on the transition from radial glia to glioblasts). This difference in abundance is most likely the consequence of a later transition from radial glia to glioblasts[1] in the anterior brain, allowing for a longer period of neurogenesis.

The multiome data allowed us to impute gene expression across the dataset, providing a direct comparison between gene expression, gene accessibility and the enrichment of TF-binding motifs. Notably, most marker genes had concordant expression and gene accessibility, with some exceptions such as *RBFOX3*, which is expressed in most glutamatergic neurons, but had low accessibility. Conversely, *TNC* was expressed only in glioblasts, but accessible in most neural cell types (Fig. 1f). We combined conventional motif discovery with gene expression for each cell type to limit identified TF motifs to those coinciding with TF expression, discarding unexpressed redundant motifs. The identified motifs included expected early neuronal (EBF1), pan-glial (SOX9) and oligodendrocyte lineage (SOX10) markers, as well as TFs with strong lineage-specific expression (for example, LHX6 and DLX2 in interneurons derived from the medial ganglionic eminence and the lateral or caudal ganglionic eminence, EMX2 in telencephalic glutamatergic neurons and OTX2 in midbrain GABAergic neurons; Fig. 1f; additional TFs in Extended Data Fig. 5). Among non-neural cells, RUNX1—which is indispensable for microglia and haematopoietic stem cell development in mice[14]—was specific to immune cells (mainly microglia and border-associated macrophages). Similarly, FOXF2, which is required for pericyte development in mice, was specific to pericytes and endothelia, whereas FOXC1 was also active in meningeal fibroblasts and vascular lepotomeningeal cells and is required for the development of the meninges in mice[15]. These findings reinforce our confidence in the identity of the main clusters.

We observed a significant 10% increase in the number of accessible regions along the neuronal differentiation trajectory, but no significant increase in the glioblast lineage ($P < 0.05$; Extended Data Fig. 4g). Indeed, in oligodendrocytes (part of the glioblast lineage), a shift towards heterochromatin has been observed in which large numbers of neuronal and later oligodendrocyte progenitor cell genes are silenced during differentiation[16,17]. Notably, the number of accessible regions also increased with age across all classes except radial glia ($P < 0.001$; coefficient 3,206; s.e. 634; $t = 5.06$; 6 d.f.; linear regression), with the newly acquired accessible regions being strongly enriched for NFI-binding sites (Fig. 2a,b). Similarly, in the combined radial glia and glioblast classes, we found that more mature cell neighbourhoods, representing primarily glioblasts, were also most associated with NFI-binding sites (Milopy; Extended Data Fig. 4d–f). Together, these features represent a general maturation function for NFI factors across different neural progeny in the developing brain, where they have been described to promote a loss of stemness[18].

## *Cis*-regulatory element specificity

We next investigated how cell-type specificity compared between chromatin accessibility and gene expression. We used the variance between the cluster-level Pearson residuals as a measure of specificity.

For most marker genes, gene expression was more specific than the sum of linked accessible regions (Extended Data Fig. 6a). By contrast, individual accessible marker regions were generally more cell-type specific than marker genes (Fig. 2c). As a consequence, we found 1,361 marker genes, but 120,183 marker regions (Fig. 2d). Thus, cCREs discovered here provide a rich source of regulatory elements with precise cell-type, cell-state and temporal resolution during brain development.

We next assessed the region specificity of accessible regions by comparing them to known functional central nervous system enhancers from the VISTA developmental enhancers database[19]. Nearly all of the VISTA enhancers overlapped with accessible regions in our data (96% overlapping feature set; 39% intergenic, 53% intronic, 4% promoter). Many VISTA enhancers are specific to the forebrain, midbrain or hindbrain, and these showed a similar pattern of activity in the scATAC-seq dataset (Extended Data Fig. 3c). In many cases these enhancers (HS; *Homo sapien* sequence) were accessible only in more specific cellular lineages such as hindbrain glutamatergic neurons (HS161; Extended Data Fig. 3d), immature interneurons in the ganglionic eminences (HS702) or radial glia and GABAergic neurons in the midbrain (HS830).

To better understand the gene-regulatory programs underlying the dataset, we identified accessible region topics using pycisTopic, which uses a latent Dirichlet allocation model to identify groups of accessible regions that covary and are likely to represent biological programs. Each cluster was downsampled to 1,000 nuclei, and we fitted a model with 175 topics on the basis of the point where the log-likelihood estimation and topic coherence scores reached saturation. A *t*-SNE plot of the accessible regions based on the topic scores showed distinct clusters linked to individual topics (Fig. 2e), representing distinct regulatory programmes. In contrast to distal elements, most TSS regions were not strongly linked to individual topics and clustered together on the embedding, indicating that they were less variable and represent constitutively open promoters. A subset of promoter-proximal regions clustered separately, and represented two topics of pan-neuronal and glial CTCF-binding sites (Fig. 2f,g). CTCF is a key factor in the establishment of genomic organization and CTCF deregulation has been shown to be involved in several neurodevelopmental disorders[20].

We used the Genomic Regions Enrichment for Annotation Tool to link topics to known biological processes through the biological annotation of nearby genes (Supplementary Table 3). For example, topics 4 and 25 were enriched for genes relevant to GABAergic interneuron identity and oligodendrocyte differentiation, respectively. When scoring the associated signatures (accessible regions in the topic related to the pathway), clear enrichments in the immature interneuron and oligodendrocyte precursor populations could be identified (Fig. 2h), respectively. As individual topics reflected only region accessibility and not gene expression, we identified enriched TF motifs for each topic and reduced them to a set of archetypal motifs (arche-motifs)[21]. This prevented the prioritization of false-positive motifs based on the similarity of the binding motif in TF families. Indeed, topic 4 was enriched for the MEIS (that is, *MEIS2*), HD/2 (that is, *DLX2* or *DLX5*), Ebox/CAGATGG (that is, *NEUROD1*) and NFI (that is, *NFIA*, *NFIB* or *NFIX*) arche-motifs, whereas topic 25 was primarily enriched for the SOX/4 (that is, *SOX10*) arche-motif (Fig. 2i).

In conclusion, we found that although accessibility gene scores were not as descriptive of cellular identify as gene expression, individual genomic regions were often highly specific and descriptive of cellular programs when analysed as coherent topics.

## Enhancer logic in neuronal specification

Although topic modelling can be a useful tool to understand the activity of accessible regions, it does not offer any explanations as to the underlying logic that drives activity of regions between cellular lineages. To better understand the syntax of regulatory elements that differentiate neuronal lineages, we trained a convolutional neural network (CNN)

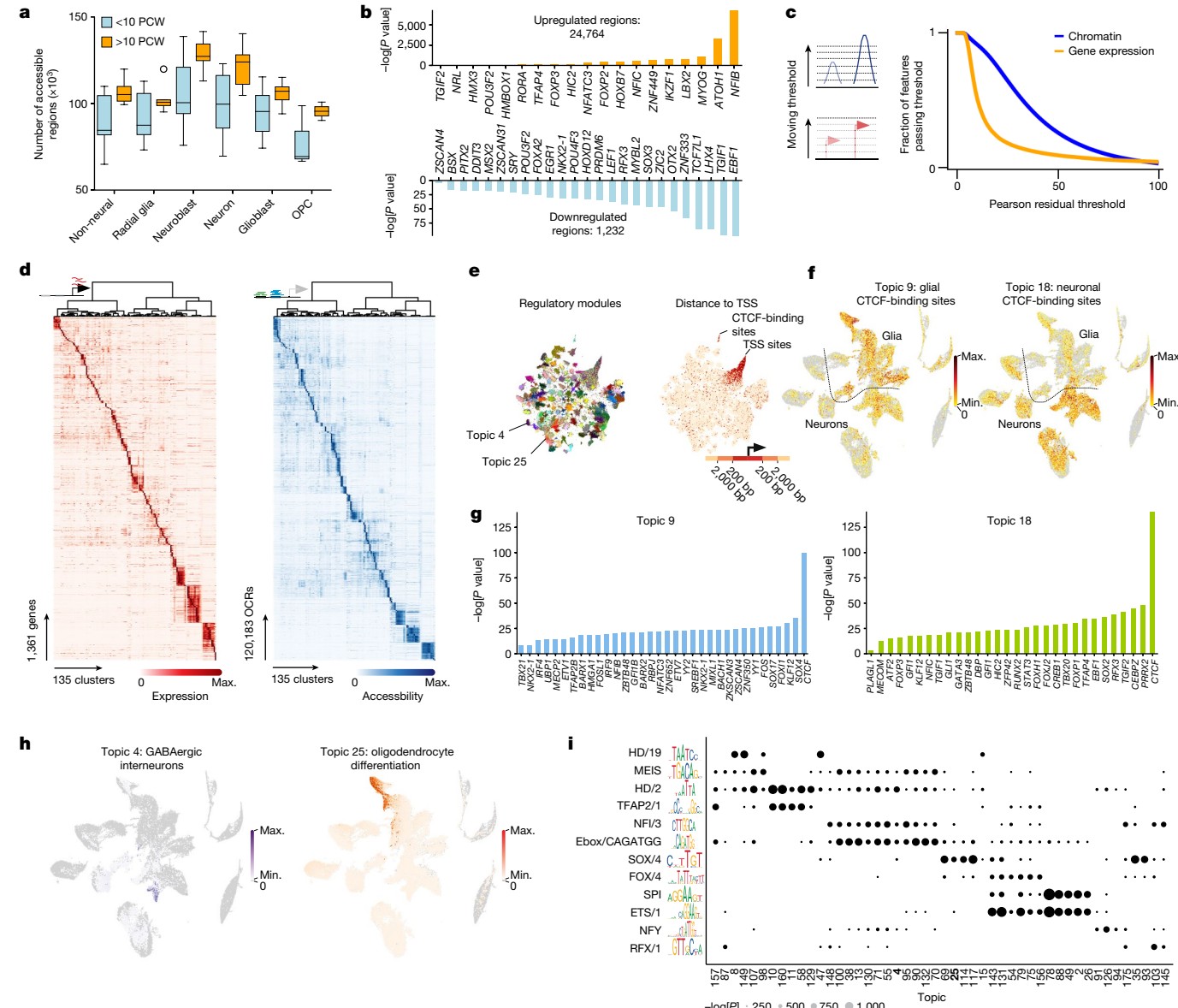

**Fig. 2 | Functional annotation of open chromatin regions. a**, Number of accessible regions by cell class, split between biological samples. $n = 26$ biologically independent samples (17 samples at <10 post-conception weeks and 9 samples >10 post-conception weeks). Box plots are centred on the median, the box represents the first to third quartiles and the whiskers extend to the minima and maxima with a maximum of 1.5× the interquartile range; points beyond this range are plotted as outliers. **b**, TF motifs in regions that are differentially accessible early or late in the dataset ($P < 0.05$; Benjamini–Hochberg-corrected one-sided Fisher exact test). **c**, Enrichment comparison between the gene expression and chromatin accessibility components of the dataset. A moving threshold was used to identify the fraction of features enriched in at least one cluster at different levels of stringency. **d**, Selection of marker gene expression and accessible marker regions. Accessible regions are limited to top 2,000 per cluster. OCRs, open chromatin regions. **e**, t-SNE plot in which dots represent accessible regions and are coloured by highest-scoring

topic. A strong enrichment of promoter regions in the top right of the second plot shows constitutively active elements. **f**, t-SNE plots of topics 9 and 18. Both were enriched for CTCF-binding sites and grouped together on the t-SNE map. **g**, Enriched TF-binding motifs in topics 9 and 18 identified using HOMER (one-sided; no multiple test correction). **h**, t-SNE plots of nuclei showing enrichment of Gene Ontology signatures enriched in topics 4 and 25 (shown in **e**), identified using the Genomic Regions Enrichment for Annotation Tool. **i**, Arche-motif enrichment for a subset of topics. Dot size represents enrichment (identified using HOMER; one-sided; no multiple test correction). The arche-motifs contain binding sites for the following TFs (non-exhaustive)−HD/19: OTX1/2; MEIS: MEIS2/3; HD/2: HOXA2/LBX2; TFAP2/1: TFAP2A/B; NFI/3: NFIA/C; Ebox/CAGATGG: PTF1A/NEUROD1/2/ATOH1; SOX/4: SOX4/10; FOX/4: FOXA1/2/FOXP2; SPI: SPI1/SPIB/C; ETS/1: ELF1/3/5/GABPA; NFY: NFYA/B/C; RFX/1: RFX1/2/3/4.

to predict cell-type identity on the basis of sequence composition[22,23]. We focused on five large, well-sampled clades: GABAergic neurons from the midbrain, glutamatergic neurons from the hindbrain and telencephalon, and granule and Purkinje neurons from the cerebellum. The model consisted of four convolutional layers followed by two dense layers and was able to predict the correct class with an average receiver operating characteristic area under curve (ROC AUC) score of 0.92

across the classes (Fig. 3a,b). We determined the contribution of each nucleotide in the target sequences towards the prediction (contribution score) using DeepExplainer, and identified short motifs with high predictive power (seqlets) that recur in the target sequences by clustering them using TF-MoDisCo. In this way, we discovered on average 6 seqlets per accessible region, and 84% of the selected regions were associated with at least one seqlet (online data; GitHub repository).

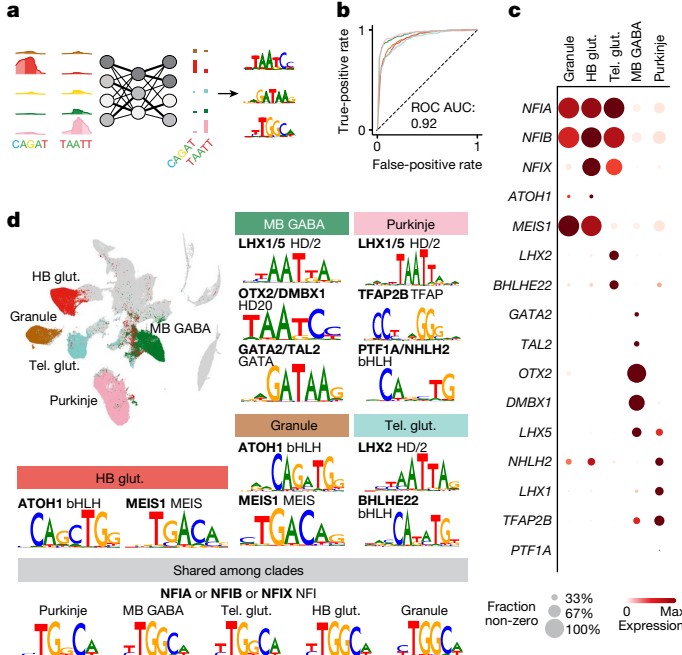

**Fig. 3 | CNN predicts neuron type from sequences. a**, Network layout. Enriched sequences for each cell type are one-hot-encoded and used to train a CNN network with four convolutional layers and two dense layers. The loadings of the convolutional layers are used to calculate the contribution of each of the nucleotides in the original sequence and recurrent patterns can be clustered to find driving TF motifs. **b**, ROC curves for each of the classes and mean ROC AUC of all classes. In a random classification model, the ROC AUC is 0.5. **c**, Dot plot representing expression of each of the TFs in **d**. Dot size represents the fraction of non-zero nuclei and the colour represents the expression level. HB, hindbrain; glut., glutamatergic; Tel., telencephalon; MB, midbrain. **d**, *t*-SNE map showing the selected classes. For each of the classes a subset of the characteristic motifs is shown. The identity of motifs was determined by the binding motif and expression of the corresponding TF in the corresponding class.

We identified *MEIS1* and *ATOH1* as key regulators in both hindbrain glutamatergic neurons and cerebellar granule neurons even though *ATOH1* expression could be detected only in a subset of the nuclei in those populations (Fig. 3c,d). Telencephalic glutamatergic neurons, in contrast, were very distinct from their posterior counterparts and were characterized by LHX2 and BHLHE22 motifs. For the GABAergic neurons, the GATA2 motif was observed only in midbrain neurons, whereas the OTX2 motif could also be seen in the Purkinje neurons, in which the gene is not expressed. This most likely represents DMBX1, a TF from the same family that is expressed at low levels in Purkinje progenitor cells both in this dataset and a previously published human neurodevelopment scRNA-seq dataset[1] (Extended Data Fig. 7a). Both populations contained the motif for TFAP2B and LHX1 or LHX5, with *LHX1* being only minimally expressed in the midbrain neurons.

## Gene regulatory dynamics in Purkinje neurons

The CNN itself did not provide a temporal order of what stages of the cell trajectory these TFs are active in. To further investigate the temporal relationship between TF expression and cCRE accessibility, we focused on the Purkinje lineage, which was well sampled in our dataset. Purkinje neurons are born in the ventricular zone of the hindbrain from *PTF1A*+ progenitors. From there they migrate into the developing cerebellum forming a characteristic layer of large, arborated neurons. We fitted a pseudotime trajectory to the 71,947 nuclei of the Purkinje lineage (Fig. 4a). We next applied DELAY, a different CNN method that

exploits the temporal shift between the expression of TFs and their targets in single-cell lineages in combination with TF-binding-site information derived from chromatin immunoprecipitation with sequencing to estimate gene regulatory networks. This revealed a network of 148 TFs co-regulating each other during Purkinje cell differentiation (Fig. 4b; network in Supplementary Table 4).

We used the inferred gene regulatory network to computationally model single nuclei using BoolODE, a tool that allows conversion of boolean TF networks to ordinary differential equation networks, recapitulating in silico the expression dynamics of TFs along the trajectory (Fig. 4c,d). One of the central TFs in the network dynamics was *ESRRB*, an oestrogen-related nuclear receptor TF that in the cerebellum is expressed uniquely in Purkinje neurons. Expression of *ESRRB* was preceded by that of a series of other TFs (*PTF1A*, *ASCL1* and *NEUROG2* in the progenitor phase; *NHLH1, NHLH2, TFAP2B, LHX5* and *PAX2* in the neuroblast phase) and itself preceded the expression of later Purkinje markers such as *PCP4*. We identified nine cCREs linked to *ESRRB*, which showed two distinct activation patterns, early and late (Fig. 4e,f). Using the CNN we had previously trained to distinguish neuronal cell types, we then identified the nucleotides driving the Purkinje lineage identity in these two groups of cCREs. We found several TFAP2B-binding motifs in the early cCREs, and an increase of LHX5-binding motifs in the late cCREs (Fig. 4g; other cCRES in Extended Data Fig. 7b). Finally, once *ESRRB* was expressed, we observed increased accessibility at its downstream binding sites elsewhere in the genome (Fig. 4e, bottom). The activation of *ESRRB* can thus be seen as a two-step process in which the gene is first poised for expression by TFAP2B, after which LHX5 binds the late cCREs and *ESRRB* expression is induced, leading eventually to the activation of ESRRB target genes. Our dataset provides rich resources—RNA expression for every TF (online data; CATlas), predicted cCREs and their activities (online data; CATlas) and predicted seqlets for every accessible region included in training the CNN (online data; GitHub)—to explore similar regulatory processes for many other genes and lineages.

## Cell type specificity of GWAS polymorphisms

Mutations in non-coding gene-regulatory regions have been implicated in numerous psychiatric disorders[24]. In many instances these non-coding regions are primarily active during a limited temporal window in selective cell types, which makes it difficult to identify the affected developmental processes[25]. Chromatin accessibility atlases with single-cell resolution spanning across several developmental time points can thus be an important tool in the identification of cell-type-specific vulnerabilities in complex trait disorders by providing increased selectivity[2,13]. To identify whether any of the cell types in our dataset were selectively vulnerable during development to mutations associated with psychiatric disorders, we curated a large set of phenotypes from the UK Biobank[26] as well as GWAS results from 11 psychiatric phenotypes[27–37]. We used stratified linkage disequilibrium score regression to identify cell types for which the phenotype was enriched for single nucleotide polymorphisms (SNPs) in the corresponding cell-type-specific accessible regions[38]. As all of the accessible regions in our dataset are of brain tissue during early development, we wanted to ensure that cell-type enrichments for a phenotype remained significant when conditioned on other life stages and tissues. We therefore added accessible regions identified throughout development[4] and adulthood[39] to the background dataset to correct for our fetal neural-focused selection of features.

We found the expected associations for many of the non-neural cell types (Extended Data Fig. 8) and several significant enrichments for the psychiatric phenotypes in neuronal subtypes. After correcting for multiple testing (Bonferroni or false discovery rate; Fig. 5a), no significant enrichments were found for Tourette's syndrome, obsessive compulsive disorder, bipolar disorder, alcohol use disorder or

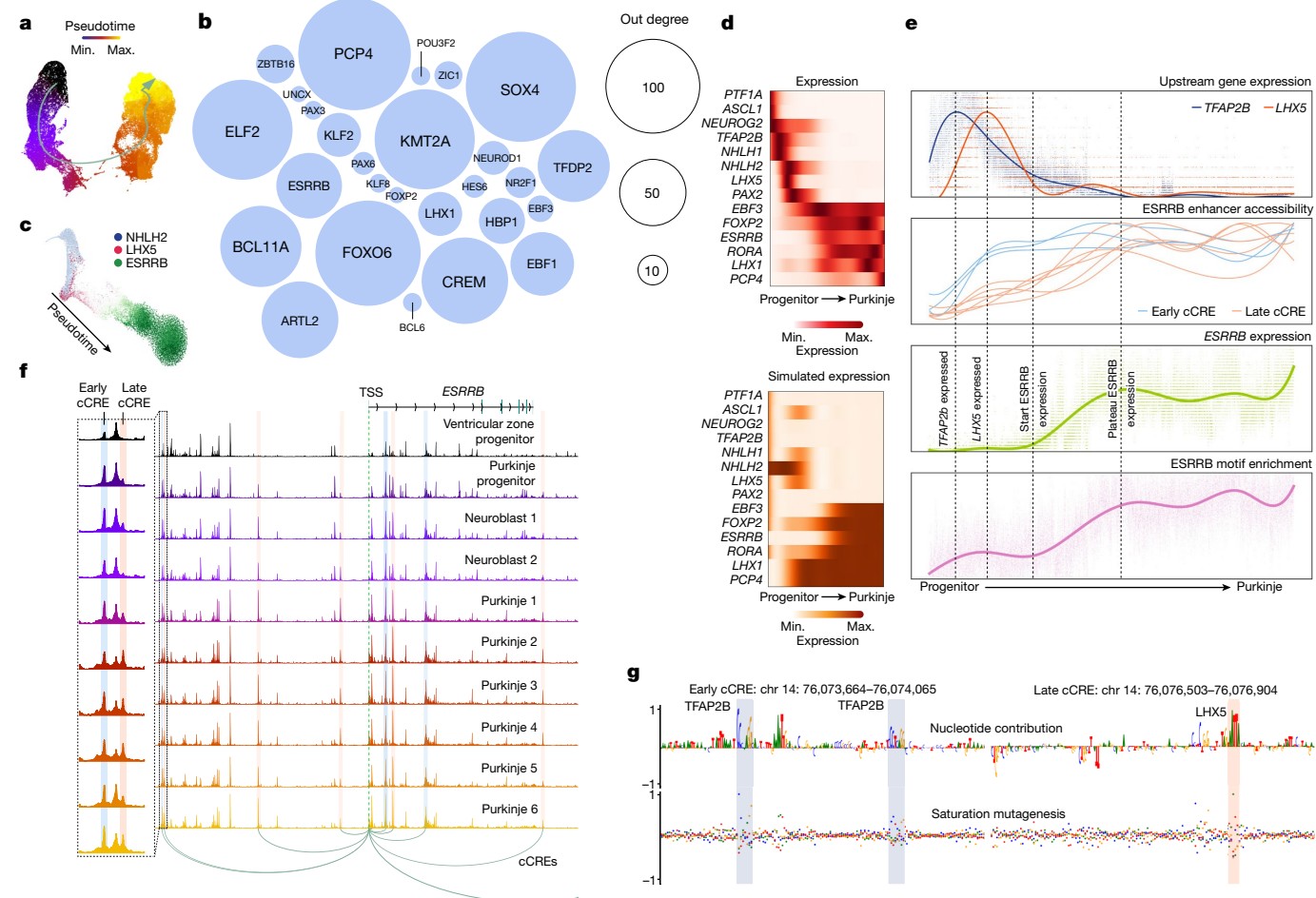

**Fig. 4 | Gene regulatory dynamics in Purkinje neurons. a**, *t*-SNE map of Purkinje neurons showing pseudotime and the differentiation trajectory. **b**, TFs involved in the gene regulatory network as identified using DELAY. The nodes are sized by centrality. **c**, *t*-SNE map of nuclei simulated from the DELAY network using BoolODE. Colour represents the expression of *NHLH2*, *LHX5* and *ESRRB*. **d**, Heat maps of Purkinje marker genes expressed along the pseudotime trajectory and the corresponding prediction based on the DELAY network. **e**, Trend lines of important factors in *ESRRB* gene progression. The vertical lines mark important events. Top to bottom: expression of *LHX5* and *TFAP2B*; accessibility of cCREs regulating *ESRRB*; expression of the *ESRRB* gene; and enrichment of the ESRRB-binding site in target peaks. **f**, Chromatin accessibility landscape around the *ESRRB* gene. Nine cCREs were identified to regulate *ESRRB* expression. Two are highlighted that occur close to each other but open up at different stages in differentiation. **g**, The contribution scores of two cCREs regulating *ESRRB* expression (an early and a late cCRE), corresponding to the marked peaks in **f**. The top plot shows the contribution score (importance of nucleotide for prediction); the bottom plot shows the effect of mutating each nucleotide variant of a region on the prediction score (saturation mutagenesis). The early example contains two binding sites for TFAP2B, whereas the late cCRE contains an LHX5-binding site.

Alzheimer's disease, although we did see lower uncorrected *P* values (Extended Data Fig. 9a; 0.003 < *P* < 0.01; Methods) in all immune cells for Alzheimer's disease compared to the neural cell types (all *P* > 0.1; Methods), which agrees with previous findings linking SNPs to immune genes[40].

Several disorders showed associations that agree with known disease biology. Schizophrenia was associated with cortical interneurons derived from the medial ganglionic eminence and *SATB2*-expressing telencephalic excitatory neurons, supporting a cortical developmental origin of the disease[35]. Attention-deficit hyperactivity disorder was associated with immature GABAergic neurons and Purkinje neuroblasts in the cerebellum, which might be related to the structural abnormalities in the cerebellum often observed in patients with attention-deficit hyperactivity disorder[41]. Anorexia nervosa was associated with interneurons derived from the lateral and caudal ganglionic eminences, in agreement with known eating-disorder associated SNPs in GABAergic receptors[42]. Autism spectrum disorder was associated with neuroblasts from the hindbrain, supporting potential involvement of the brainstem in autism spectrum disorder[43]. For insomnia,

*TAL2*-expressing GABAergic neurons in the midbrain were implicated, in line with the reported role of such neurons in the reticular formation of the ventral midbrain in wakefulness[44,45].

The strongest associations, however, were those observed between midbrain-derived GABAergic neurons (several groups) and major depressive disorder (MDD), which we validated in a second cohort[46] (Supplementary Table 5; stratified linkage disequilibrium score regression; one-sided; Benjamini–Hochberg $\alpha = 3.37 \times 10^{-5}$). The involvement of GABAergic neurons in MDD is well established[47], but often attributed to cortical interneurons for which we found no significant associations. Midbrain GABAergic neurons, however, are also known to be involved in the regulation of reward behaviour and stress[48], two systems known to be disrupted in MDD. Moreover, a subset of these *SOX14*-expressing midbrain-derived neurons also migrate to the thalamus and pons[5], suggesting a potentially broader effect from these mutations. The overlap between MDD and insomnia in *TAL2*-expressing midbrain GABAergic neurons is also notable as the two disorders have high comorbidity[49].

To better understand the association between MDD and midbrain GABAergic neurons, we used cCREs to identify target genes in MDD. We

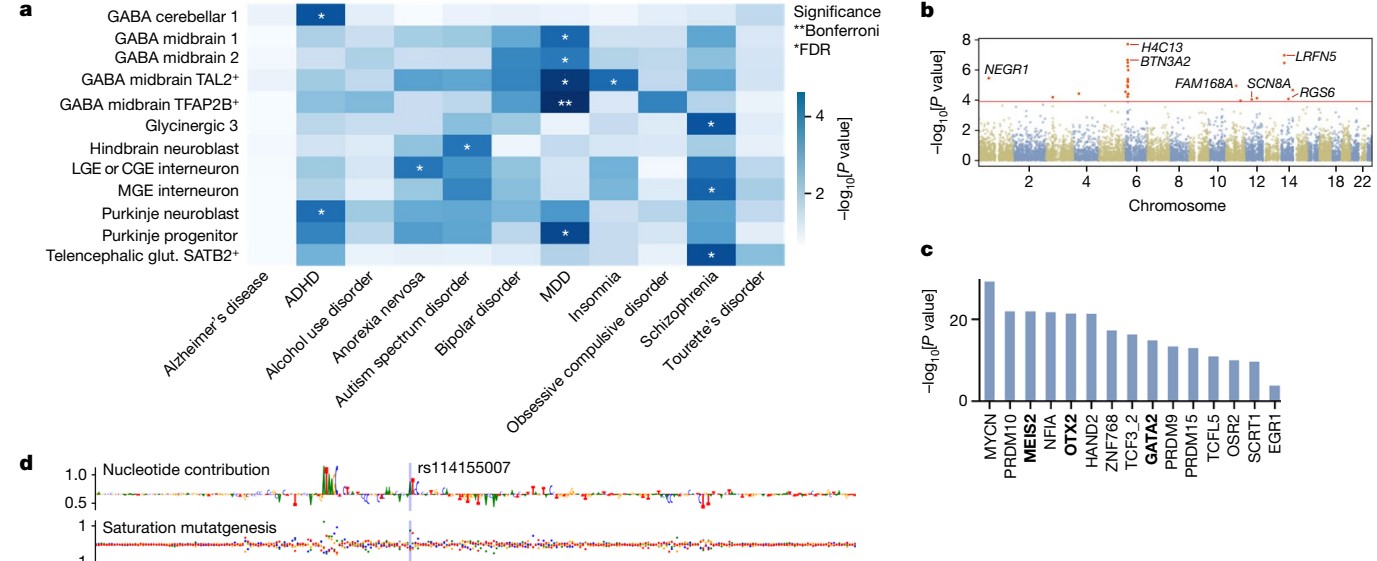

**Fig. 5 | Enrichment of psychiatric SNPs in first-trimester central nervous system cell types. a**, $-\log_{10}[P$ values] for neuropsychiatric phenotypes derived from stratified linkage disequilibrium score regression analysis (one-sided). Asterisks indicate significant (FDR or Bonferroni) differences. Only cell types reaching significance in one of the phenotypes are plotted. ADHD, attention-deficit hyperactivity disorder. CGE, caudal ganglionic eminence; LGE, lateral ganglionic eminence; MGE, medial ganglionic eminence. **b**, MAGMA $-\log_{10}[P$ values] for gene-associated cCREs with the Benjamini–Hochberg $\alpha$ set at $1.2 \times 10^4$. **c**, Enriched TF-binding motifs in the MDD-associated SNPs passing Bonferroni correction (HOMER; one-sided; no multiple test correction). MEIS2, OTX2 and GATA2 are TFs strongly associated with midbrain inhibitory neurons. **d**, The contribution scores of region chr 6: 28,885,244–28,885,645, which contains rs114155007, one of the SNPs associated with MDD.

pooled the set of cCREs linked to each individual gene and used MAGMA to identify genes significantly associated with MDD. This yielded 25 associated genes consisting mostly of known MDD genes such as *NEGR1*, *BTN3A2*, *LRFN5* and *SCN8A*, as well as a number of histone genes located in the same locus as *BTN3A2* (Fig. 5b; *H2AC13*, *H2AC15*, *H2BC14*, *H2BC15* and *H4C13*). Although many of these genes were expressed in midbrain GABAergic neurons, none was specific to these neurons. Conversely, accessible regions significantly associated with MDD were enriched for the MEIS2, OTX2 and GATA2-binding motifs, indicating a midbrain GABAergic identity (Fig. 5c and Extended Data Fig. 9b), with 29 of the 114 significant MDD regions also containing CNN-predicted OTX2-binding sites and 46 containing predicted GATA2-binding sites. We also identified binding motifs from *MYCN* and *PRDM10*, which are expressed broadly in the developing brain, and *NFIA*, which had higher expression levels in glutamatergic neurons and glioblasts. Midbrain GABAergic neurons are unlikely to be the sole contributor to MDD aetiology, with the other cell types perhaps being more affected during adulthood or later stages of development. For instance, a similar methodology has been used to link intratelencephalic-projecting neurons in the adult brain to MDD[13] and excitatory hippocampal neurons have also been linked to MDD[50].

We next examined individual accessible regions and the predicted nucleotide contributions to the midbrain GABAergic fate (DeepEx-plainer scores). For most MDD-associated SNPs, we did not find immediately interpretable overlaps, with only rs114155007 directly overlapping with the OTX2-binding site (Fig. 5d). We do not expect the effects of these SNPs in midbrain GABAergic neurons to be primarily mediated through the disruption of key cell-fate-defining TFs. In conclusion, these findings suggest that some broadly expressed genes associated with MDD contribute to disease when perturbed specifically in midbrain GABAergic neurons during early neurodevelopment.

## Discussion

In this study we provide a high-resolution multiomic atlas of chromatin accessibility and gene expression in the first-trimester human brain. We identified more than 100,000 cell-type- and region-specific developmental accessible chromatin regions, inferred cCREs and predicted their regulatory syntax using CNN modelling. These resources enable analyses that span from developmental lineages to individual nucleotides—linking TFs to putative enhancers, and enhancers to their target genes—as exemplified here by our analysis of the regulation of *ESRRB* in the Purkinje neuron lineage.

Our dataset further enabled analysis of genetic association with disease. We found that most genes linked to MDD were not cell-type specific, yet the associated accessible regions showed enriched TF motifs consistent with midbrain GABAergic neurons. This suggests that dysregulation of those genes contributes to MDD only when the dysregulation affects specific midbrain cell types (but may cause other phenotypes when dysregulated in other cell types). The observation reinforces the fact that disease-associated alleles are contextual, and yield disease phenotypes mainly by their effect in specific cell types. Nonetheless, our GWAS analysis covered only a relatively early period of neurodevelopment and more complete datasets will be required to fully elucidate the genetics of complex diseases relative to brain cell types.

In conclusion, this study provides a rich resource for the study of early embryonic human neurodevelopment in the context of gene regulation and neurodevelopmental disease.

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

## Methods

### Sample collection

All experiments in this study followed all relevant guidelines and regulations, including the International Society for Stem Cell Research 2021 guidelines. Human fetal samples were collected from routine termination of pregnancies at the Karolinska University Hospital, Addenbrooke's Hospital in Cambridge and the Human Developmental Brain Resource following informed consent of the donors. The use of fetal samples collected from abortions was approved by the Swedish Ethical Review Authority and the National Board of Health and Welfare (Etikprövningsmyndigheten; DNR2020-02074). In the UK, approval from the National Research Ethics Committee East of England, Cambridge Central was obtained as well as approval from the North East – Newcastle & North Tyneside 1 Research Ethics Committee (Local Research Ethics Committee, 96/085; DNR2019-04595). The samples were dissected by a trained embryologist into the major developmental regions (telencephalon, diencephalon, mesencephalon and metencephalon) along the anterior–posterior axis. In addition, the cerebellum was separated from the metencephalon and when possible the metencephalon was divided into medulla oblongata and pons. Following the dissection, the samples were transferred to ice-cold Hibernate E medium (ThermoFisher, A1247601) and either shipped overnight at refrigerated temperature to Sweden or processed the same day when collected at the Karolinska University Hospital.

Some important limitations of this study must be considered. First, as these are clinical samples, the timing was variable and based on expert annotation rather than knowledge of the date of conception. Second, owing to damage incurred during collection, not all regions could be collected from every sample and had to be compensated for by collecting more samples. Third, as the samples were derived from several sources the time between collection and dissociation varied.

### Statistics and reproducibility

Sample size was dictated by the availability of scarce early developmental human samples, and based on previous experience with similar studies in mice. No power calculations were carried out to determine sample size. The investigators were not blinded, as this was an exploratory study with anonymous untreated samples. The reproducibility of the dataset across specimens was assessed by the contribution of donors to each cluster (Extended Data Fig. 1d). The linkage disequilibrium score regression analysis linking MDD to midbrain GABAergic neurons was validated in a separate GWAS cohort (see below).

### Nucleus isolation

Tissue was gently minced using a razor blade and incubated with the papain dissociation system (Worthington) following the manufacturer's recommendations (including 200 U ml⁻¹ DNAse), at 37 °C for 10 min. The suspension was then triturated using glass pipettes to dissolve any remaining chunks of tissue, before being filtered through a 30-µm filter (CellTrics). The cells were then washed with EBSS, concentrated (200$g$, 5 min) and counted using a haemocytometer, after which $1 \times 10^6$ cells were pelleted (500$g$, 5 min) in a 2-ml LoBind Eppendorf tube and pelleted. The cell pellets were dissociated for 5 min on ice using 100 µl of dissociation mix (0.001% digitonin, 0.01% Non-idet P40, 1 mM dithiothreitol, 1 U µl⁻¹ RNAse inhibitor, 0.1% Tween-20, 1% BSA, 10 mM Tris-HCl, 10 mM NaCl, 3 mM MgCl$_2$). When only scATAC-seq was carried out, no RNAse inhibitor or dithiothreitol was added to the mix. Dissociation was halted by addition of 1 ml of wash buffer, after which nuclei were pelleted again (500$g$, 5 min) and resuspended in 1× nuclei buffer (10x Genomics) and recounted.

### Single-cell sequencing

Libraries were generated using the 10x Genomics Chromium Controller and Single Cell ATAC or Single Cell Multiome ATAC + Gene Expression kits. Briefly, a targeted number of nuclei (5,000–10,000) was treated with a Tn5 transposase for 60 min at 37 °C to fragment the DNA and insert adapter sequences into open parts of the chromatin. The suspension was then mixed with the provided barcoding PCR mix and a gel-bead emulsion was generated by co-encapsulating the suspension with barcoded beads in the 10x microfluidic chip and PCR with reverse transcription was carried out in a C1000 Touch thermal cycler (Bio-Rad) with one of two programs—ATAC: 12 cycles of (5 min at 72 °C, 30 s at 98 °C, 10 s at 98 °C, 30 s at 59 °C, 1 min at 72 °C) and hold at 15 °C; or multiome: 45 min at 37 °C, 30 min at 25 °C and hold at 4 °C. For multiome samples, quenching agent was added to prevent the PCR with reverse transcription reaction from continuing. Following PCR, the DNA was isolated from the droplets and cleaned up with Cleanup mix and silane Dynabeads. Sample indices and P7 primers (Illumina) were ligated during library construction using the following PCR protocol: 9 or 10 cycles of (45 s at 98 °C, 20 s at 98 °C, 30 s at 67 °C, 20 s at 72 °C) and 1 min at 72 °C before holding at 4 °C. SPRIselect beads were used for size selection of fragments to generate the final library. The fragment size distribution was analysed using the Bioanalyzer high-sensitivity chip to eliminate libraries that did not show the expected nuclear banding pattern. Libraries were then sequenced using the Illumina Nova-seq instrument using the recommended setting for paired-end sequencing, with the scATAC-seq and scATAC-seq (multiome) libraries in separate flow cells as pooling of them is not recommended with a target of 100,000 read pairs per nucleus. The multiome scRNA-seq libraries were pooled with other 10x Genomics scRNA-seq v3.1 libraries.

### 10x data processing

All samples were demultiplexed and aligned to the human genome GRCh38.p13 Gencode v35 primary sequence assembly using either Cellranger-atac 2.0.0 or Cellranger-arc 2.0.0 for scATAC-seq and single-cell multiome, respectively. The RNA libraries from multiome samples were aligned as described previously[1].

### Chromograph pipeline

Chromograph is a new analysis pipeline for scATAC-seq data based on the key architecture of Cytograph 2 (ref. 8), which uses loom files as the underlying data format and is available for use in GitHub (https://github.com/linnarsson-lab/chromograph). The results in this paper were generated using commit #9ae1434. In brief, chromograph provides tools to pool and split scATAC-seq data, carry out clustering, carry out balanced peak calling based on cluster partitions, and identify marker peaks and enriched TF motifs, and enables imputation of gene expression from limited multiome data. This dataset was analysed by first carrying out a primary analysis, and then manually splitting it into subsets based on marker genes. These subsets were then reanalysed, and the results were again pooled to generate a more fine-grained dataset than the primary analysis. The pybedtools and pybigwig packages were used to work with bed files and bigwig files, the loompy package was used to work with loom files, numpy was used to work with matrices and numba was used to speed up computations wherever possible. Scikit-learn and statsmodel were used for more complex calculations and pynndescent was used for fast nearest-neighbour computations.

### scATAC-seq quality control

TSS enrichment was calculated using pycisTopic[51] (TSS window 50 base pairs (bp), flanking window 1,000 bp) as we noticed discernible change in some of the samples after updating Cellranger-arc. Samples with a score below 5 were discarded. For the other samples, nucleus-by-bin matrices were generated at both 5-kb and 20-kb resolution with bins that overlapped with any of the ENCODE blacklist[52] being removed. The 5-kb nucleus-by-bin matrix was used for doublet detection using an adapted version of DoubletFinder. In brief, nuclei were co-embedded with 20% artificial doublets to determine a threshold to distinguish doublets from singlets on the basis of their nearest-neighbour network

and a doublet score was assigned on the basis of each nucleus's local neighbourhood. For the multiome samples, the RNA-doublet score was used as it proved slightly more stable. Additionally, the sex of the sample was determined on the basis of the fraction of Y-chromosomal reads (>0.05% for male) as well TSS fraction. Nuclei that were not doublets and had more than 5,000 and fewer than 100,000 fragments, more than 20% TSS fragments and more than 1,000 RNA unique molecular identifiers (UMIs), and at least 10% unspliced RNA UMIs were pooled to generate the main dataset (the final two filters apply only to multiome).

On average 27,599 high-quality fragments per nucleus were identified with a fragments in peaks ratio of 54%. High-quality nuclei were selected on the basis of the number of fragments and the fraction of fragments overlapping TSS as well as UMI count and splice ratio in multiome samples.

## Preclustering and consensus peak calling

The feature set used to generate the nucleus-by-peak matrices is dynamically derived from the data through peak calling. To do so, the 20-kb matrices were first joined and binarized, after which the top 20% of autosomal bins were selected with an upper threshold of 60% coverage across the dataset and decomposed using latent semantic indexing (LSI; more detailed description below). A $k$-nearest-neighbour graph was then constructed, and the data was clustered into broad clusters using Louvain clustering. Fragments from the nuclei belonging to each cluster were then aggregated and randomly split in two to generate two pseudobulk replicates per cluster. The pseudobulk aggregates were then downsampled to 25 million fragments and MACS2[53] was used to call peaks using the following parameters: callpeak -f BEDPE -g hs --nomodel --shift 100 --ext 200 --qval 5 × 10$^{-2}$ -B −SPMR. Peaks were then extended to 400 bp using BEDtools and non-overlapping peaks between the pseudo-replicates were discarded. Next the identified peaks for all clusters were pooled and clustered using BEDtools cluster. For each cluster of peaks, the centre point was extracted and extended to 400 bp to generate the consensus peak set. Peaks overlapping with the ENCODE blacklist were removed and the remainder were annotated using HOMER[54] on the basis of Gencode v32, after which the nucleus-by-peak matrix was generated.

## Latent semantic indexing

Decomposition was carried out in two steps. First the matrix was depth-normalized and infrequent features were upweighted by carrying out a term frequency–inverse document frequency transformation. The resulting non-binary matrix was then used to compute the principal components using an incremental principal component analysis. Initially 40 components were computed, but components that are not distributed significantly differently from their predecessor are discarded along with a depth-correlated component if present. Next the components were batch-corrected using Harmony to mediate chemistry and sample effects[55].

## Clustering, embedding and aggregation

The nucleus-by-peak matrix was decomposed using an iterative LSI, meaning that the data was decomposed and clustered in two rounds. First the top 20,000 features by total coverage from the autosomal chromosomes were used to carry out preclustering, after which 20,000 autosomal features were selected again on the basis of the variance of their precluster-level enrichment for a second LSI. Batch effects were again corrected for using Harmony. The second LSI is then used to generate nearest-neighbour graphs and carry out Louvain clustering. A $t$-SNE map was then generated using an adapted version of 'the art of using $t$-SNE'[56] that better preserves global structure than native $t$-SNE. Additionally, a uniform manifold approximation and projection was generated using UMAP-learn[57] with default settings. For both methods, Euclidean distances were used as a metric. Next all clusters were aggregated and a normalized counts per million layer was added.

The enrichment of individual peaks was calculated as a Pearson residual[58]. In brief, fragments were modelled as a negative binomial distribution for which the expected accessibility is the product of the total number of fragments per cluster ($c$) and the fraction of fragments per peak ($g$). The residuals can then be calculated as the difference between the observed ($X$) and expected ($\hat{\mu}$) accessibility corrected by the negative binomial variation (dispersion parameter fixed at 100 for all analysis in this paper). For each cluster, the top 2,000 peaks by Pearson residual were marked as marker peaks. The 20,000 peaks with most variance between Pearson residuals were used to calculate cluster similarities and to generate the cluster dendrogram.

$$Z_{cg} = \frac{X_{cg} - \hat{\mu}_{cg}}{\sqrt{\hat{\mu}_{cg} - \frac{\hat{\mu}_{cg}^2}{\theta}}} \quad \theta = 100$$

## Gene expression imputation and marker selection

Thirty-one percent of nuclei in the dataset were processed using the Single Cell Multiome ATAC + Gene Expression kit. This allows for the imputation of gene expression measurements in the ATAC-alone samples. To predict gene expression in the scATAC-seq nuclei, first all multiome nuclei were scaled to 5,000 UMIs and an 'anchor' net was generated consisting of a directed graph of each scATAC-seq nucleus and their 10 nearest multiome neighbours. Next the weights were scaled to sum to 1 for each nucleus and the nearest-neighbour matrix was multiplied by the gene expression profiles of the multiome nuclei to generate predicted gene expression profiles for each scATAC-seq nucleus.

Trinarization scores and gene enrichment were calculated as defined previously[2] with marker genes being selected on the basis of their enrichment. The trinarization scores were then used for auto-annotation using a set of punch cards specific to early human development[1,4].

## Subset analysis and pooling

The dataset was split on the basis of cluster-level marker expression into the following partitions: fibroblast (*DCN* and *COL1A*-expressing), immune (*PTPRC*-expressing), vascular (*TAGLN*-expressing or *CLDN5* and *FLT1*-expressing), oligodendrocyte progenitor cell (*PDGFRA* and *OLIG1*-expressing), radial glia or glioblast (*HES1*-expressing or *BCAN* and *TNC*-expressing) and neuronal (expressing any of *INA*, *NHLH1*, *GAD2*, *SLC17A6* or *SLC6A5*) lineages. The subsets were reanalysed using the same pipeline described above. Clusters that contained fewer than 10 multiome nuclei or for which less than 1% of the total cluster size was multiome nuclei were excluded as well as clear clusters of doublets. The neuronal lineage partition was split for a second round into GABAergic (*GAD2*), glutamatergic (*SLC17A6*, *SLC17A7* or *SLC17A8*) and peptidergic lineages. All partitions were then pooled again and new summary statistics and embeddings were generated.

## Motif enrichment

For every cluster, the 2,000 selected marker peaks were used as input to HOMER findMotifsGenome[54] using GC-matched genomic sequences as background. The Hocomoco v11 Full collection was chosen as the TF-binding motifs to be tested. The naming convention was manually altered to reflect genes names in the gene expression analysis. This allowed the filtering of false positives by exclusion of nucleus–motif combinations for which the corresponding TF was unlikely to be expressed (trinarization score < 0.5). Additionally, all TFs were assigned to a family on the basis of their arche-motif[21].

## Gene accessibility and cCREs

Gene-accessibility scores were computed using an adapted version of the cicero workflow[59] using the python SKGGM package. First, the distance parameter was estimated by optimizing the calculation of the

regularized covariance matrices for 100 random 500-kb regions. Next the distance-adjusted covariance for each accessible region with each TSS site was calculated in 500-kb bins with a 250-kb overlap. Most pairs are sampled twice and pairs with inconsistent covariances are discarded (about 5%). The co-accessibility cutoff was set empirically by testing the number of subnetworks over varying cutoff thresholds. Gene activity scores were then calculated by multiplying the peak-by-nucleus and region-to-TSS covariance matrices, normalizing against size factors derived from a linear regression model and pooling across the 25 nearest neighbours. Similarly to the region–TSS covariance matrix, cCREs were identified by calculating the region–gene expression covariance.

## Identification of total accessible regions by sample
To identify general trends in opening and closing of chromatin, all fragments from individual cell classes and biological samples were pooled together and MACS2 was used to call peaks per class per sample. A one-sided Fisher exact test (using the fisher python package) with Benjamini–Hochberg correction was used to identify differential regions. A generalized linear model was used to estimate the influence of age on the number of accessible regions.

## VISTA enhancer overlap
CNS enhancers (from the VISTA database[19]) were downloaded and lifted over to GRCh38 using UCSC liftOver, excluding any that could not be confidently lifted over, resulting in 620 enhancers, of which 596 overlapped with our peak set. The enhancers that were specific to the forebrain, midbrain and hindbrain according to the original authors were isolated (total of 159, 75 and 78, respectively), the corresponding peaks in the dataset were identified and the brain region with the highest accessibility was identified, after which the Jaccard similarity was calculated.

## PycisTopic modelling
The full dataset was downsampled to a maximum of 10,000 nuclei per cluster to reduce computational burden and prevent over-representation. The number of topics was varied from 25 to 500 at intervals of 25, running for 50 iterations with an $\alpha$ of 50 divided by the number of topics and a $\beta$ of 0.1. The most stable model (175 topics) was selected on the basis of topic coherence and log-likelihood in the last iteration. The region-topic scores were normalized so that they summed to 1 for every nucleus and a $t$-SNE map was generated for the regions and binarized topic lists were generated by assigning each region to the topic that it scored the highest on. Next each topic was used as input for HOMER with the Hocomoco TFs and the results were reduced to the highest-scoring representative of each arche-motif group. The binarized topics were also used as input for Genomic Regions Enrichment for Annotation Tool analysis[60] to identify Gene Ontology terms describing each topic. For some selected terms, the associated regions (in the topic) were used to calculate an enrichment score using the signature_enrichment function of pycisTopic[51].

## Enhancer CNN
Nuclei from all clusters annotated as Purkinje, midbrain GABA, cerebellum granular neuroprogenitor, hindbrain glutamatergic or telencephalic glutamatergic were grouped into five superclusters and enrichment between the clusters was recalculated and peaks were included for learning only if the log-fold change with the second highest accessibility was more than 1. One-hot-encoded sequences (401 bp) were used as input to a CNN trained as a classification model using pyTorch. The network consists of 4 convolutional layers of 256, 60, 60 and 120 nodes and kernel sizes 7, 3, 5 and 3, respectively, and each layer was followed by batch normalization, RELU activation and maximum pooling. There were then 2 dense layers of 256 nodes with batch normalization, RELU activation and a dropout rate of 0.4. A softmax normalization was applied to the final output layer and cross-entropy

loss was used as the loss function with label smoothing set to 0.1. The model was trained using an Adam optimizer with a learning rate of 0.01. The model was trained for 26 epochs.

Contribution scores for each sequence were calculated using DeepLiftShap's (deepExplainer[61]) attribute function using the mean of the input sequence shuffled 100 times as background. The hypothetical score was calculated for each possible nucleotide in the sequence by multiplying the contribution by the background-corrected input[62]. TF-MoDisCo[62] was then applied to all of the sequences enriched in a cluster with a flanking size of 5 bp, a sliding window of 15 bp and a minimum cluster size of 30 seqlets.

## Pseudotime, generalized additive models and ChromVAR
For analysis of the Purkinje lineage, all clusters labelled 'Purkinje' and the *PTF1A*-expressing cluster of ventricular zone progenitors were isolated and a new $t$-SNE map was generated. pySlingshot was then used to calculate the pseudotime. pyGAM was used to fit gene and cCRE trends to the Purkinje neuron lineage with gene expression being modelled using a Poisson generalized additive model and cCRE accessibility using a linear generalized additive model. ChromVAR was applied using the JASPAR human PWM (human_pwms_v2) to compute motif variability.

## Supervised inference and stochastic simulation of Purkinje gene regulatory network
We used DELAY[63] (https://github.com/calebclayreagor/DELAY) to infer the Purkinje gene regulatory network from gene-accessibility dynamics in pseudotime and then carried out stochastic simulations to verify the putative network's gene-expression dynamics. First, we retrained DELAY on a large scATAC-seq dataset of plasma B cell differentiation data[64] with ground-truth data from chromatin immunoprecipitation with sequencing[65] to prepare the neural network to infer the Purkinje gene regulatory network from tens of thousands of single nuclei. Then, we fine-tuned DELAY on the Purkinje developmental trajectory using ground-truth targets of a cerebellar ataxia-related gene, ataxin 7 (ref. 66). For the final gene regulatory network inference, we used the expression-linked, log-normalized gene-linked peak counts from all TFs that were differentially expressed in at least 1% of Purkinje nuclei across pseudotime (Supplementary Tables 6 and 7; TradeSeq; Wald test; two-sided). We then used BoolODE[67] to simulate the expression of each gene in the network given its top eight most likely regulators.

## GWAS enrichment
Accessible region locations were lifted over to GRCh37. Features were binarized on the cluster level with a Pearson residual threshold of 10. Cluster heritability was calculated using linkage disequilibrium score regression[38]. As a background, we used the merger of our feature set with the features from development[4] and adulthood[39]. Only SNPs from hapmap3 were included to reduce imputation errors. In total we tested 325 phenotypes from the UK Biobank[26] and 11 psychiatric phenotypes[27–37]. All used UK Biobank phenotypes had non-zero heritability estimates ($z$ score > 4). Results for UK Biobank phenotype enrichments were corrected for the number of cell types using FDR or Benjamini–Hochberg procedures. For the psychiatric enrichments, FDR and Bonferroni corrections were applied for the number of cell types and tests ($\alpha = 3.37 \times 10^{-5}$, 135 × 11 tests).

Two different MAGMA[68] tests were conducted with default settings. First, the cCREs linked to genes were annotated to genes in a custom MAGMA annotation file. A MAGMA gene analysis was used to assess which genes were affected in MDD. Next, MAGMA gene analyses were conducted for ADHD, anorexia, autism spectrum disorder, MDD and schizophrenia, on a custom annotation file in which individual accessible regions were treated like individual genes to identify specific deregulated elements. Accessible regions passing Benjamini–Hochberg correction were then used as input for HOMER with the full vertebrate motif reference.

## Reporting summary

Further information on research design is available in the Nature Portfolio Reporting Summary linked to this article.

## Data availability

Raw sequencing data are available through from the European Genome Phenome Archive (EGAS00001007472). To facilitate ease of use of the resource, the chromatin accessibility and gene expression data are browsable through the CATlas web browser (http://catlas.org/human-braindev) and the CNN and anonymized cell-ranger outputs can be downloaded through GitHub at https://github.com/linnarsson-lab/fetal_brain_multiomics[69].

## Code availability

All code used to reproduce the figures is available through GitHub at https://github.com/linnarsson-lab/fetal_brain_multiomics[69]. Code to reanalyse the data is available through GitHub at https://github.com/linnarsson-lab/chromograph[69]. The DELAY models trained on scATAC-seq data are available through GitHub at https://github.com/calebclayreagor/DELAY[69].

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

**Acknowledgements** We thank the women who have decided to donate to science, making this study possible; the Developmental Tissue Bank (Department of Neurobiology, Care Sciences and Society, Karolinska Institutet) core facility, the Human Developmental Biology Resource (UK) and Cambridge University for providing prenatal tissue samples; Y. E. Li and B. Ren for hosting the online dataset; J. B. Munting for providing supplementary code used while training the CNN; E. Vinsland for assistance collecting samples; I. Kapustova for providing illustrations; and all members of the Linnarsson laboratory and M. Bartosovic, M. Kabbe and J. Hjerling-Leffler for discussions. This research was supported by the National Institute for Health and Care Research Cambridge Biomedical Research Centre (BRC-1215-20014). The views expressed are those of the authors and not necessarily those of the National Institute for Health and Care Research or the Department of Health and Social Care. This work is financed by the following grants: an Erling-Persson Foundation Human Developmental Cell Atlas grant (S.L.), Knut and Alice Wallenberg Foundation grants 2015.0041, 2018.0172 and 2018.0220 (S.L.), Swedish Foundation for Strategic Research SB16-0065 (S.L.) and EU Horizon2020 BRAINTIME project 874606 (S.L.). We also acknowledge support from the National Genomics Infrastructure in Stockholm financed by the Science for Life Laboratory, the Knut and Alice Wallenberg Foundation and the Swedish Research Council. For the purpose of open access, the author has applied a Creative Commons Attribution (CC BY) licence to any author accepted manuscript version arising from this submission.

**Author contributions** C.C.A.M. and S.L. designed the experiments in this study. C.C.A.M. and L.H. carried out experiments. C.C.A.M. analysed and visualized the data. P.L. contributed to bioinformatics analysis. C.C.R. analysed the Purkinje neuron TF network. M.S. and D.P. carried out stratified GWAS. X.L., X.H., R.A.B. and E.S. contributed to the sample collection. C.C.A.M. and S.L. wrote the manuscript with contributions from all authors.

**Funding** Open access funding provided by Karolinska Institute.

**Competing interests** S.L. is a paid scientific adviser to Moleculent, Combigene and the Oslo University Center of Excellence in Immunotherapy. All other authors declare no competing interests.

## Additional information

**Correspondence and requests for materials** should be addressed to Sten Linnarsson.

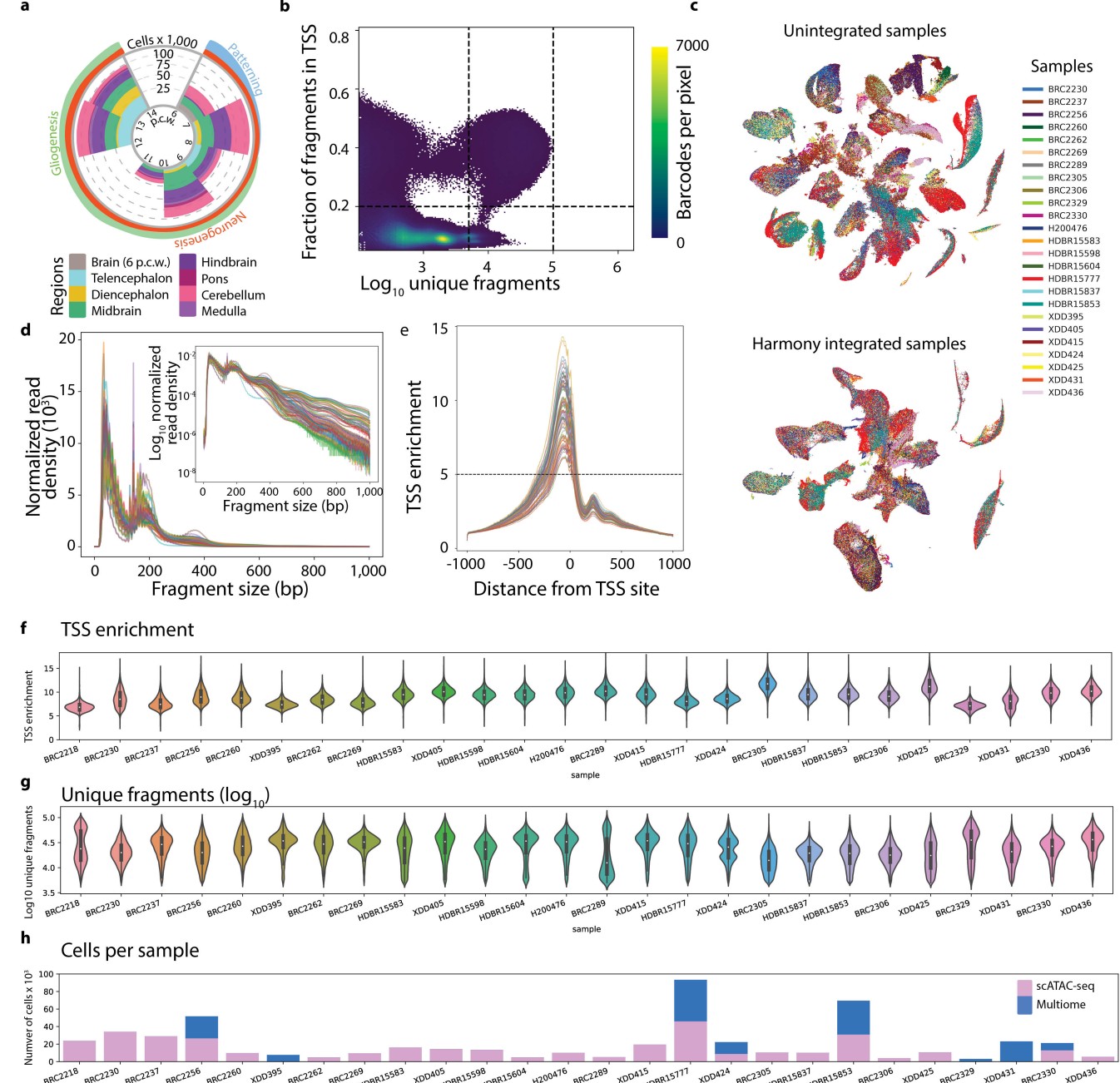

**Extended Data Fig. 1 | Quality control.** A) Collected nuclei counts per post-conceptual week (p.c.w.) and region. B) Distribution of fragment count (log10) and fraction of fragments in TSS for collected barcodes. C) t-SNE embedding generated from Latent Semantic Indexing without Harmony sample correction (top) and with sample correction (bottom). D) fragment size distribution per sample. Top plot shows log scaled density. E) TSS enrichment per sample. 5 was used as a minimum sample level cut-off. F) Distribution of TSS enrichment across nuclei per sample. n = 26 biologically independent samples

(number of cells per sample in H). Box plots within the violins are centered on the median, the box represents the first to third quartiles and the whiskers extend to the minima/maxima with a maximum of 1.5x the interquartile range, points beyond this range are plotted as outliers. G) Fragment count (log10) across nuclei per sample. n = 26 biologically independent samples. Boxplot representations follow the same rules as F. H) Number of nuclei collected per sample, separated by method.

## Data analysis workflow

## Iterative workflow

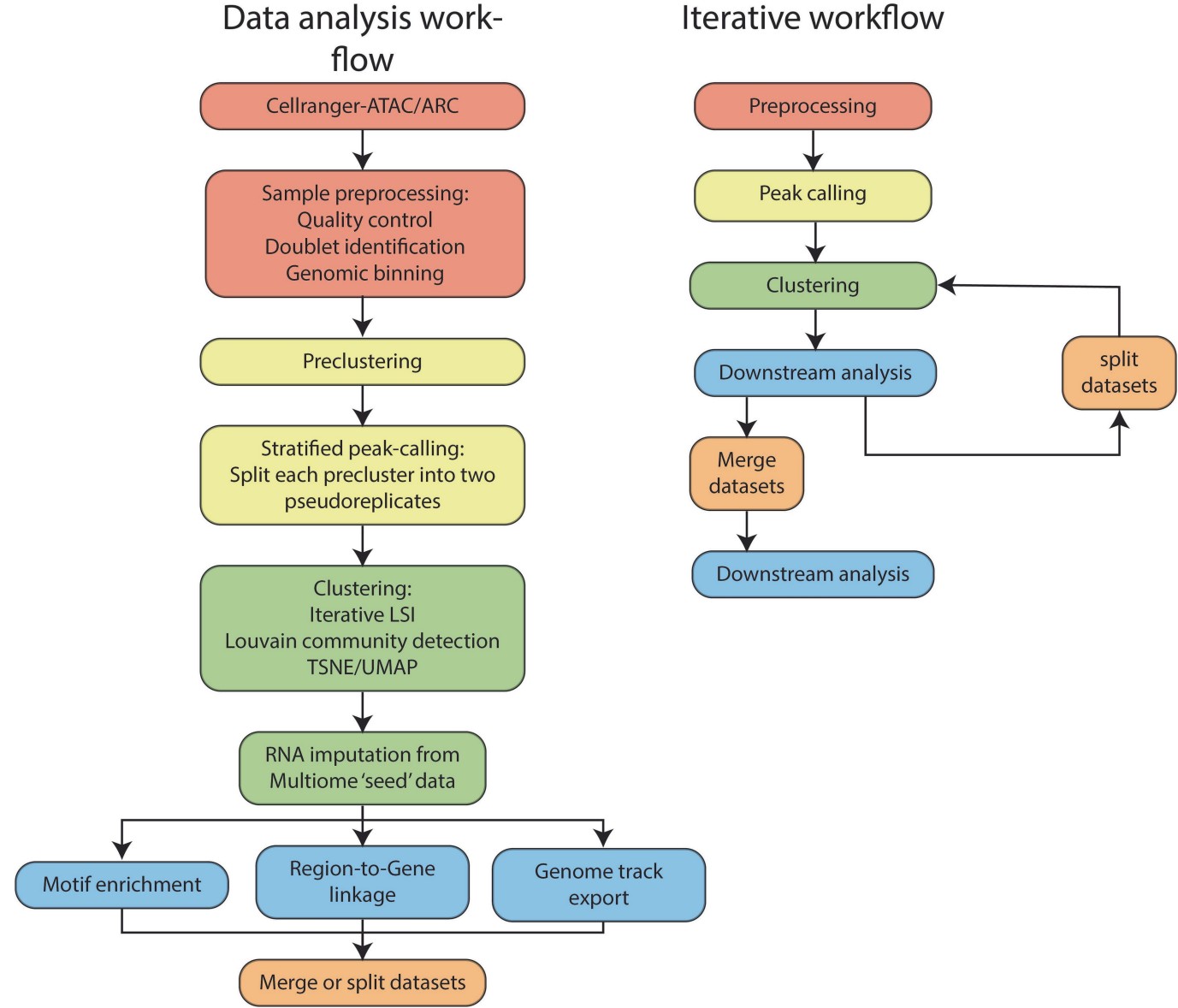

**Extended Data Fig. 2 | Analysis pipeline.** Steps taken to analyze single nucleus data. Following quality control the dataset is clustered using genomic bins as features. Peak calling is then performed per cluster and a nucleus-by-peak matrix is generated and nuclei are clustered. The available multiome nuclei are then used to impute gene expression across the dataset. Downstream analysis is performed including motif enrichment analysis and region-to-gene linkage before splitting the dataset by cell class. Each subset is reclustered and reanalyzed separately before being pooled together again using the subset clusters and a final analysis round is conducted.

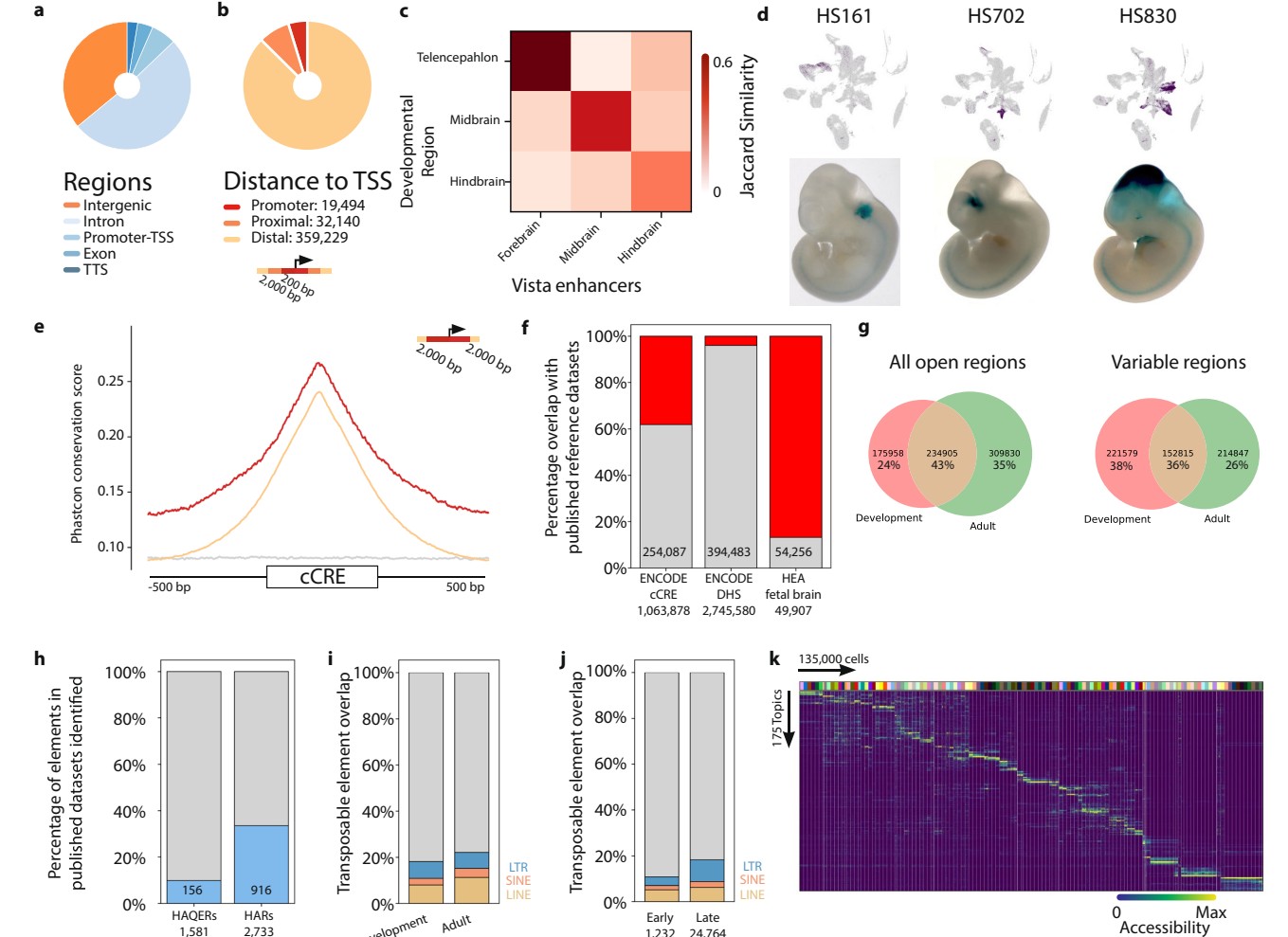

**Extended Data Fig. 3 | Annotation of accessible chromatin regions.**
A) Distribution of functional region annotations in relation to nearby genes.
B) Distribution of accessible region distance to nearest TSS. C) Jaccard
similarity between region-specific enhancers from the VISTA database and
accessible regions identified in corresponding regions of the dataset.
D) Spatially restricted accessibility of developmental enhancers overlapping
with known enhancer sequences from the VISTA enhancer-database[19] shown
by LacZ staining. From left to right, active in Hindbrain neurons and glioblasts,
immature interneurons in the Ganglionic Eminence and Midbrain radial glia
and inhibitory neurons. E) Mean DNA conservation of proximal (<2,000 bp
from TSS) and distal elements based on the PhastCon 100-way. F) Number of
accessible regions that overlap with the ENCODE cCRE and DNAse hypersensitive
site reference datasets. Additionally the number of elements that overlap
with the human enhancer atlas fetal brain dataset. Red shows regions not in
the reference dataset, gray are overlapping regions. G) Overlap between the

identified accessible regions in this study (development) and a comparable
study in the adult human brain (Li et al., 2022 under revision). The second panel
shows the overlap between variable regions in the two datasets (pearson
residuals > 10 in at least one cluster). Interestingly, a large number of regions
that are variable in development seem to be invariable in adult. H) Overlap with
two sets of evolutionarily accelerated regions, with overlap in blue and regions
from the comparison list not in our dataset in gray. Human accelerated regions
(HARs) are regions with increased rates of nucleotide substitution that are
conserved in other species, while human ancestor quickly evolved regions
(HAQERs) are regions that diverged rapidly between humans and chimpanzees
that were not previously constrained. I) Overlap with annotated transposable
elements. J) Comparison of transposable elements in early vs. late nuclei across
the dataset. K) Heatmap of region topics across the 135,00 nuclei included
in the topic modeling. Bottom images in **d** reproduced from ref. 19, https://
enhancer.lbl.gov.

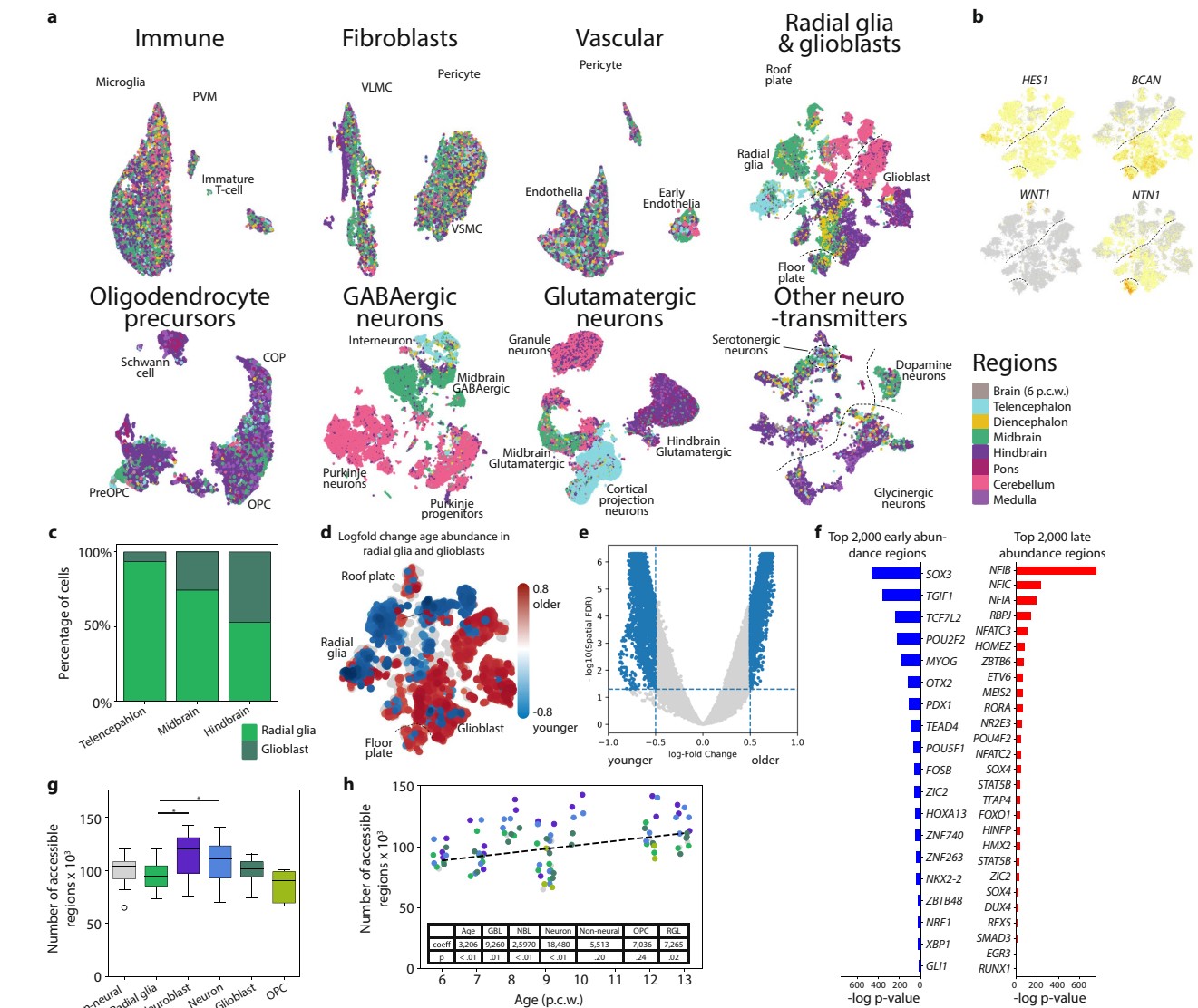

**Extended Data Fig. 4 | Regionalization of cell types.** A) Annotation of region of origin for each cell class. While clear effects of regionalization can be seen in the neural lineage, the non-neuronal nuclei are more similar across brain regions. B) Expression of canonical markers used to annotate radial glia, glioblasts, the roof plate and the floor plate. C) Distribution of radial glia and glioblasts between brain regions. D) tSNE showing change in abundance of early vs. late nuclei in local neighborhoods among radial glia and glioblasts. E) Volcano plot of change in abundance of early vs. late nuclei in local neighborhoods among radial glia and glioblasts as identified using milopy[70]. F) Enriched transcription factor motifs (HOMER; one-sided; no multiple test correction) among the 2,000 most enriched accessible regions between early and late neighborhoods (by pearson residual). G) Boxplot showing the number of accessible regions identified in different cell types. Neuroblasts and neurons have significantly more accessible regions than radial glia (two-sided independent t-test; neuroblast: t = 3.6; CI = 8,343-30,358; Cohen's D = 1.13; 38 DF; p = 0.001; neurons: t = 2.5; CI: 2,485-22,706; Cohen's D = 0.77; 43 DF; p = 0.015). n = 26 biologically independent samples, the number of cells in each class were 57,210, 160,928, 69,716, 141,189, 83,800 and 13,251 respectively. Box plots are centered on the median, the box represents the first to third quartiles and the whiskers extend to the minima/maxima with a maximum of 1.5x the interquartile range, points beyond this range are plotted as outliers. H) Linear regression fitted to age to predict number of accessible regions using the cell types as covariates (t-test; two-sided; p-values in figure).

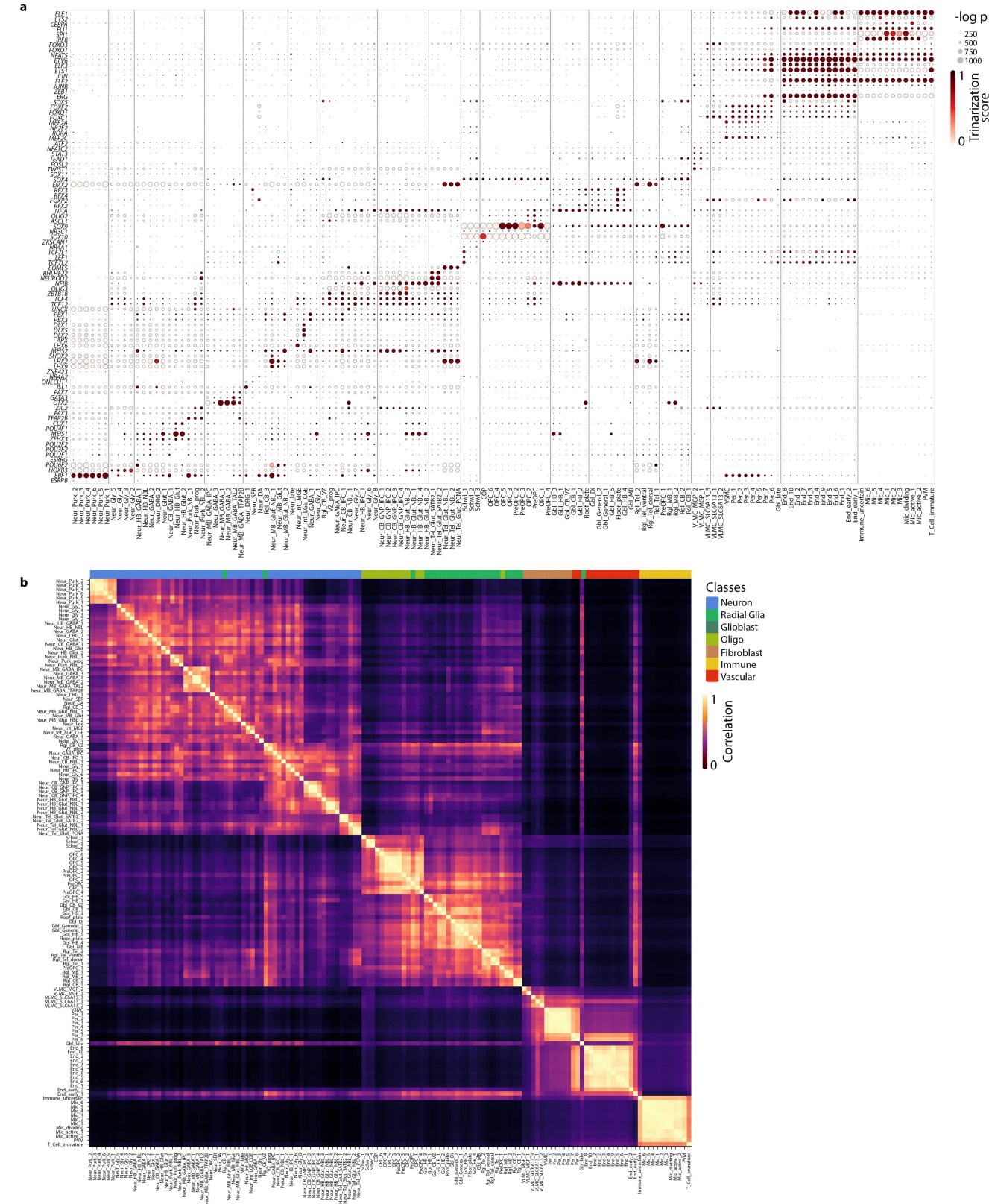

**Extended Data Fig. 5 | Cluster annotation.** A) Extended motif enrichment plot of all clusters (HOMER; one-sided; no multiple test correction). Dot size represents -log p-value of the motif enrichment. The color represents expression level. B) Correlation between cell types based on marker peak accessibility. The colorbar at the top represents the assigned cell class.

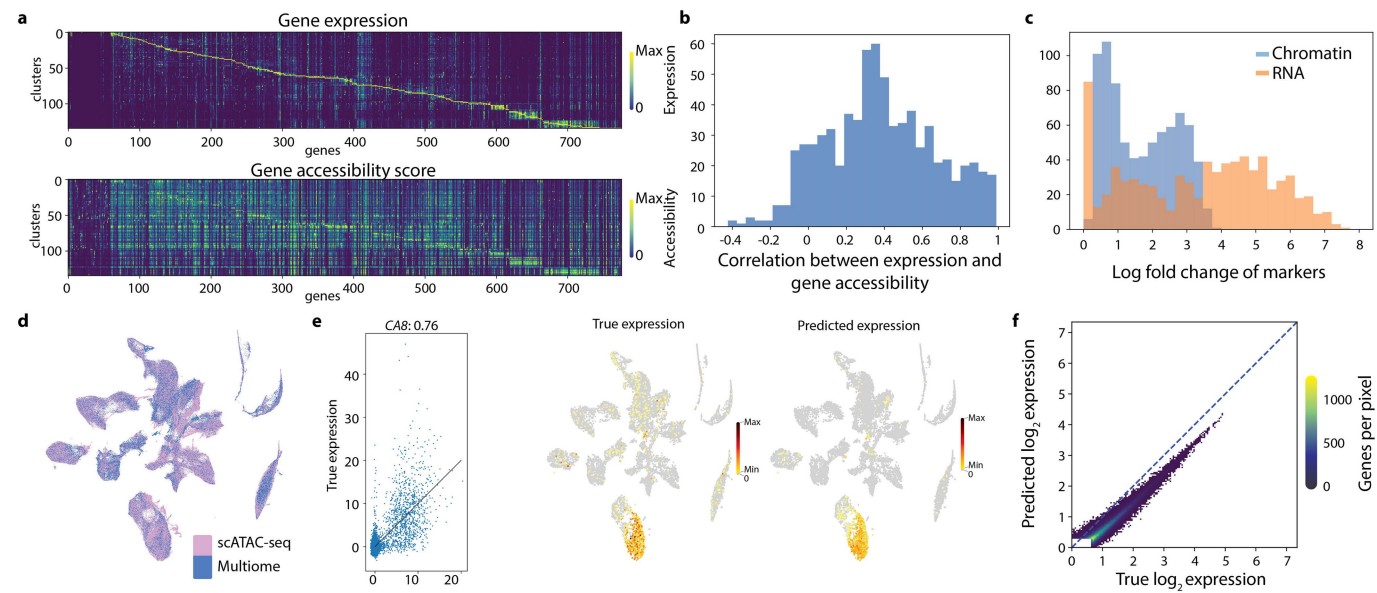

**Extended Data Fig. 6 | Gene expression imputation.** A) Comparison between cell type specificity of marker genes in gene expression and gene accessibility space. Top 5 enriched genes were selected for each cluster in both modalities and the union was used for plotting. B) Correlation between expression and accessibility of each selected marker gene. C) Maximum log fold change of marker versus median expression. The genes with RNA enrichment of zero are not expressed. D) Distribution of scATAC-seq and multiomic nuclei across the tSNE embedding. E) Leave-one-out validation of imputed expression. Predicted versus true expression of *CA8*. F) Top 100,000 gene-cluster expression pairs true vs imputed expression pairs.

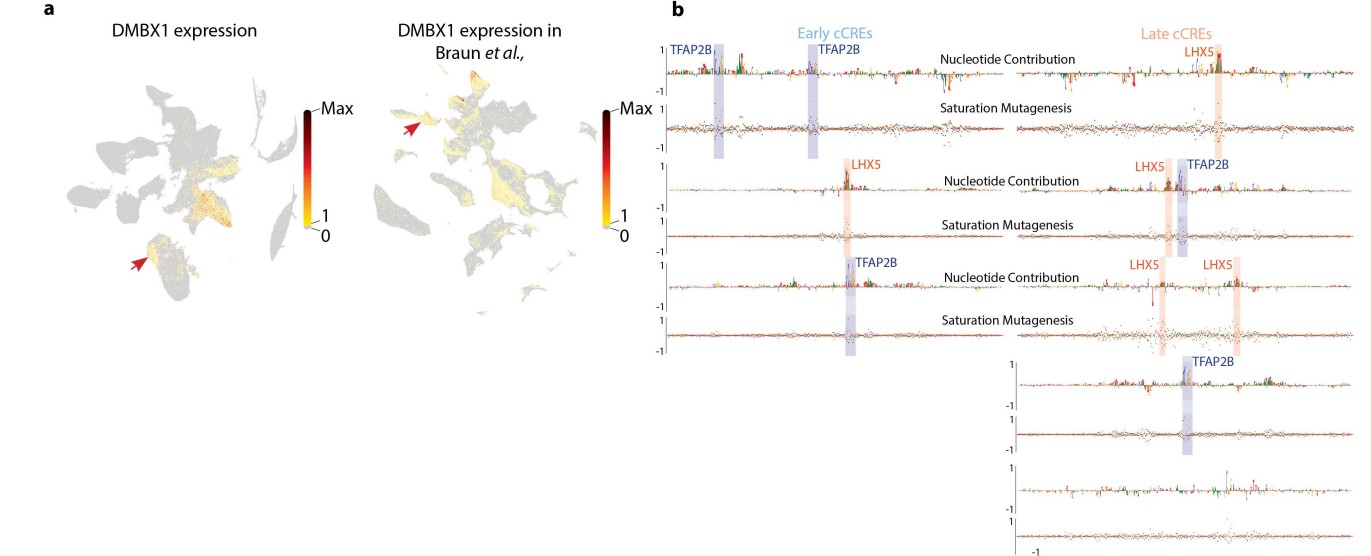

**Extended Data Fig. 7 | Additional figures related to TF CNN model.** A) Expression of *DMBX1* in this dataset and Braun et al.[1], B) Additional cCREs upstream of *ESRRB*.

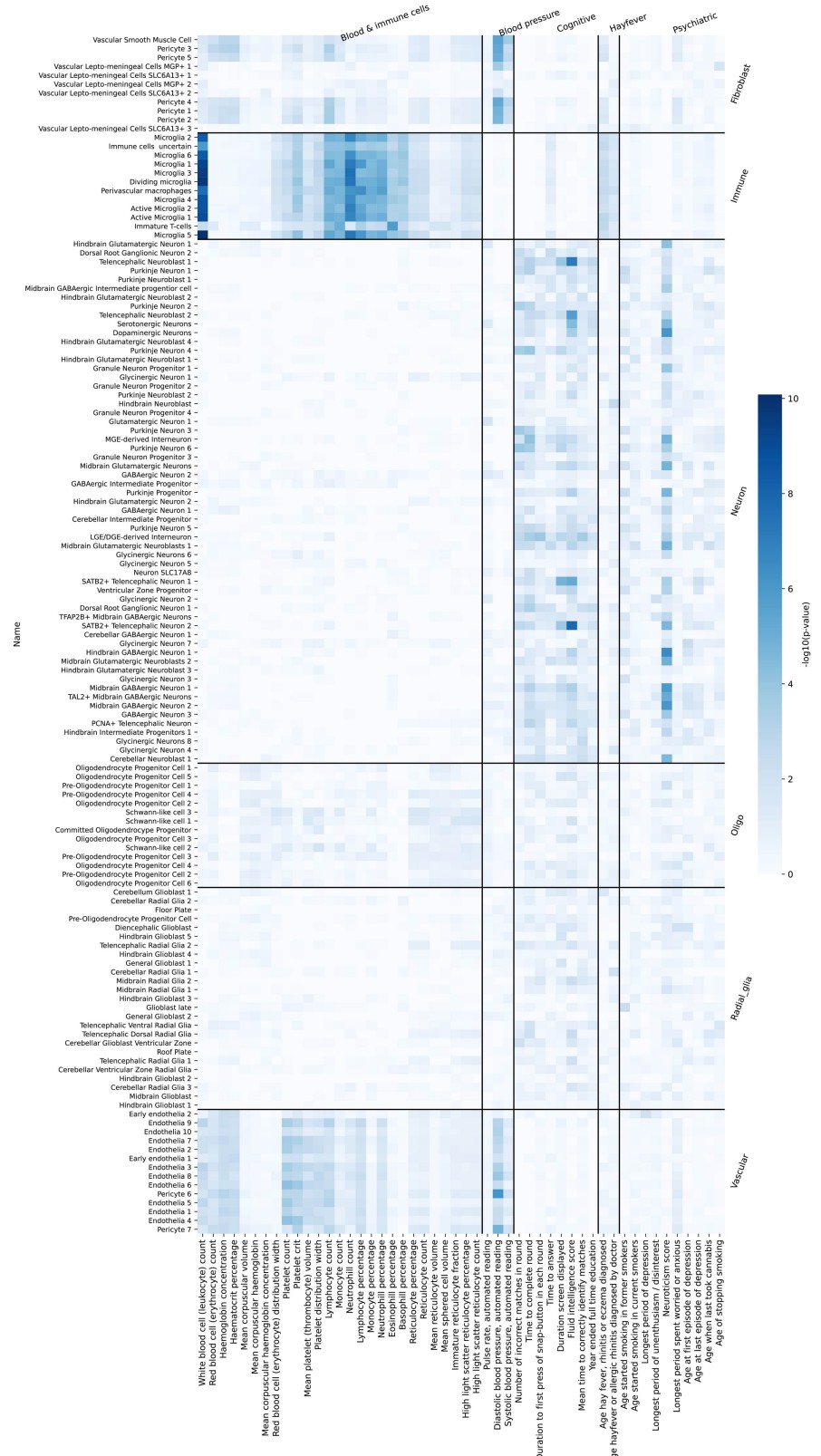

**Extended Data Fig. 8 | Enrichment of selected UK Biobank traits in neurodevelopmental cell types.** Each trait is assigned to a group of related traits (Immune, blood pressure, cognitive, hayfever and psychiatric) and LDSC analysis (one-sided) is used to identify susceptible cell types. Nuclei are ordered by major cell class. Most traits show the expected enrichment pattern.

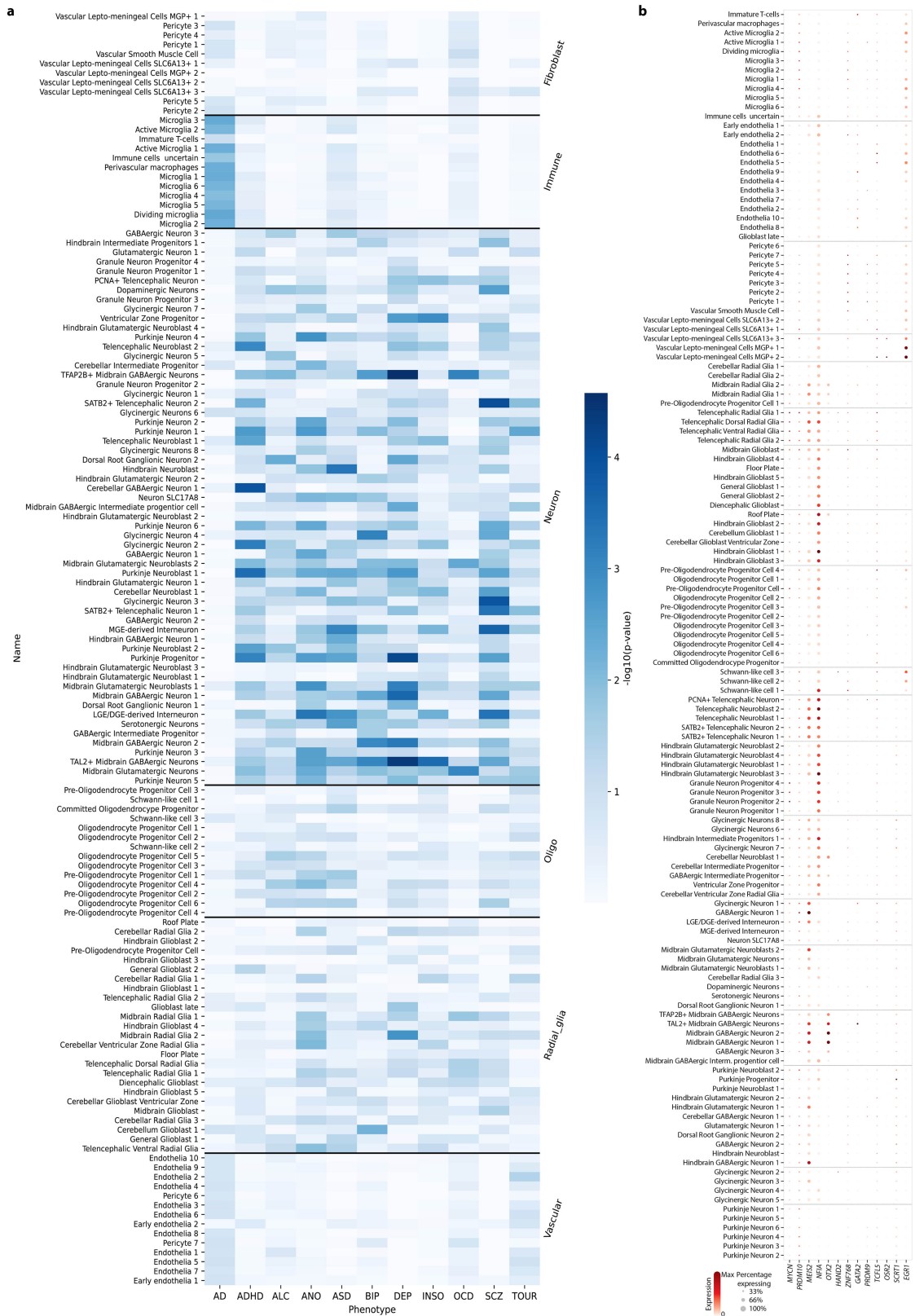

**Extended Data Fig. 9 | Enrichment of selected psychiatric phenotypes in neurodevelopmental cell types.** A) Nuclei are ordered by major cell class and LDSC analysis (one-sided) is used to identify susceptible cell types. While not reaching significance after multiple test correction, an increased association between Alzheimer's disease and immune cells can be observed in opposition to the other traits which primarily are associated with neuronal cells. B) Expression of transcription factors identified in Fig. 5c.

# Reporting Summary

## Statistics

For all statistical analyses, confirm that the following items are present in the figure legend, table legend, main text, or Methods section.

| n/a | Confirmed | |
|---|---|---|
| ☐ | ☒ | The exact sample size (*n*) for each experimental group/condition, given as a discrete number and unit of measurement |
| ☐ | ☒ | A statement on whether measurements were taken from distinct samples or whether the same sample was measured repeatedly |
| ☐ | ☒ | The statistical test(s) used AND whether they are one- or two-sided<br>*Only common tests should be described solely by name; describe more complex techniques in the Methods section.* |
| ☐ | ☒ | A description of all covariates tested |
| ☐ | ☒ | A description of any assumptions or corrections, such as tests of normality and adjustment for multiple comparisons |
| ☐ | ☒ | A full description of the statistical parameters including central tendency (e.g. means) or other basic estimates (e.g. regression coefficient) AND variation (e.g. standard deviation) or associated estimates of uncertainty (e.g. confidence intervals) |
| ☐ | ☒ | For null hypothesis testing, the test statistic (e.g. *F*, *t*, *r*) with confidence intervals, effect sizes, degrees of freedom and *P* value noted<br>*Give P values as exact values whenever suitable.* |
| ☒ | ☐ | For Bayesian analysis, information on the choice of priors and Markov chain Monte Carlo settings |
| ☐ | ☒ | For hierarchical and complex designs, identification of the appropriate level for tests and full reporting of outcomes |
| ☐ | ☒ | Estimates of effect sizes (e.g. Cohen's *d*, Pearson's *r*), indicating how they were calculated |

*Our web collection on statistics for biologists contains articles on many of the points above.*

## Software and code

Policy information about availability of computer code

| Data collection | 10X Cellranger-ATAC 2.0.0<br>10X Cellranger-ARC 2.0.0 |
|---|---|
| Data analysis | Programs:<br>Homer v4.11<br>bedtools v2.25.0<br><br>Python packages:<br>`# Name              Version          Build  Channel`<br>`cytograph          2.0.1            dev_0   <develop>`<br>`deeplift           0.6.13.0         pypi_0   pypi`<br>`fisher             0.1.9            pypi_0   pypi`<br>`harmony-pytorch    0.1.4            pypi_0   pypi`<br>`loompy             3.0.6            dev_0   <develop>`<br>`macs2              2.2.7.1          pypi_0   pypi`<br>`milopy             0.1.1            pypi_0   pypi`<br>`modisco            0.5.16.2         dev_0   <develop>`<br>`numba              0.51.0           pypi_0   pypi`<br>`numpy              1.21.6           pypi_0   pypi`<br>`opentsne           0.4.4            pypi_0   pypi`<br>`pybedtools         0.8.1            pypi_0   pypi` |

```
pybigwig            0.3.17          pypi_0  pypi
pygam               0.8.0           pypi_0  pypi
pynndescent         0.4.8           pypi_0  pypi
scikit-learn        0.23.2          pypi_0  pypi
statsmodels         0.12.1          pypi_0  pypi
ucsc-bedgraphtobigwig    377        h446ed27_1  bioconda
ucsc-bigwigaverageoverbed 377       h446ed27_1  bioconda
ucsc-liftover       447             h954228d_0  bioconda
umap-learn          0.4.6           pypi_0  pypi
pytorch             1.12.1          py3.9_cuda11.6_cudnn8.3.2_0   pytorch
pytorch-lightning   1.7.7           pypi_0  pypi
pycistopic          1.0.2.dev9+gaf3977c       dev_0   <develop>
pyslingshot         0.0.2           pypi_0  pypi
skggm               0.2.8           pypi_0  pypi

R:
ChromVAR 1.22.1

for GRN:
DELAY v0.1.0 https://github.com/calebclayreagor/DELAY
BoolODE v0.1 https://github.com/Murali-group/BoolODE

for GWAS enrichment:
LDSC v1.0.1 https://github.com/bulik/ldsc
MAGMA v1.0 https://ctg.cncr.nl/software/magma

All custom code for analysis is available through:
https://github.com/linnarsson-lab/fetal_brain_multiomics
https://github.com/linnarsson-lab/chromograph
```

For manuscripts utilizing custom algorithms or software that are central to the research but not yet described in published literature, software must be made available to editors and reviewers. We strongly encourage code deposition in a community repository (e.g. GitHub). See the Nature Portfolio guidelines for submitting code & software for further information.

# Data

Policy information about availability of data

All manuscripts must include a data availability statement. This statement should provide the following information, where applicable:

- Accession codes, unique identifiers, or web links for publicly available datasets
- A description of any restrictions on data availability
- For clinical datasets or third party data, please ensure that the statement adheres to our policy

Data Availability. Raw sequencing data is available through from the European Genome Phenome Archive (EGAS00001007472). To facilitate ease of use of the resource the chromatin accessibility and gene expression data are browsable through the CATlas webbrowser (http://catlas.org/humanbraindev) and the convolutional neural network can be downloaded through github: https://github.com/linnarsson-lab/fetal_brain_multiomics.
Code Availability. All code used to reproduce the figures is available through github: https://github.com/linnarsson-lab/fetal_brain_multiomics. Code to reanalyze the data is available through: https://github.com/linnarsson-lab/chromograph. The DELAY models trained on scATAC-seq data are available through https://github.com/calebclayreagor/DELAY.

All our data is aligned to the GRCh38.p13 gencode V35 primary sequence assembly (https://www.gencodegenes.org/human/release_35.html)
We compared our data to the VISTA database (https://enhancer.lbl.gov) and the ENCODE cCRE datasets (https://www.encodeproject.org). We also used the HOCOMOCO transcription factor PWM database (https://hocomoco11.autosome.org). For the stratified LDSC analysis we made use of the UKBiobank (http://www.ukbiobank.ac.uk) and studies from the PGC (https://pgc.unc.edu).

# Research involving human participants, their data, or biological material

Policy information about studies with human participants or human data. See also policy information about sex, gender (identity/presentation), and sexual orientation and race, ethnicity and racism.

| Reporting on sex and gender | An equal number of male and female embryos were collected (13 male, 13 female) and used for all analysis in this manuscript. To correct for structural genomic differences between the sexes, X&Y chromosomal reads were disregarded during clustering. We found no clear sex differences in cell type abundances or sex-derived artifacts in the clustering. As such the findings in this paper refer to both sexes.

As our samples were collected and processed as soon as possible, the sex of the sample could not be determined ahead of time, as such we collected from both sexes and identified the sex based on the presence of Y-chromosomal reads in the sequence data. For each cell in the data matrices the assigned sex is available as a column attribute.

Gender was not considered in this manuscript as this is generally considered a post-natal phenomenon. |
|---|---|
| Reporting on race, ethnicity, or | Samples were collected anonymously from donation in Sweden and the United Kingdom. The donation procedures did not allow us to collect race or ethnicity or similar information about the donors. As such we do not know if the ethnic |

| other socially relevant groupings | backgrounds of the samples are broadly representative of the general population, but they are likely to skew towards a western/northern European Caucasian background. |
|---|---|
| Population characteristics | Samples were collected only from healthy abortions, meaning there was no medical reason to the end the pregnancy. No chromosomal aberrations were detected in the clinic for these samples nor were any other diagnoses made. |
| Recruitment | Donors were recruited after electing to proceed with voluntary termination of pregnancy. As we do not have access to personal information of the parents, we do not know if there a self-selection bias in the data. In short, there might be differences between  different parts of the public in their propensity to donate to science. For instance, there might be a relative over-representation of education level of the mother among our samples compared to the total of performed abortions for that reason. However, we do not expect this to impact the data in a strong way. |
| Ethics oversight | For UK, by the National Research Ethics Committee East of England, Cambridge Central and the North East – Newcastle & North Tyneside 1 Research Ethics Committee (DNR2019-04595); for Sweden by Etikprövningsmyndigheten (DNR2020-02074). |

Note that full information on the approval of the study protocol must also be provided in the manuscript.

# Field-specific reporting

Please select the one below that is the best fit for your research. If you are not sure, read the appropriate sections before making your selection.

☒ Life sciences  ☐ Behavioural & social sciences  ☐ Ecological, evolutionary & environmental sciences

For a reference copy of the document with all sections, see nature.com/documents/nr-reporting-summary-flat.pdf

# Life sciences study design

All studies must disclose on these points even when the disclosure is negative.

| Sample size | Sample size was dictated by the availability of scarce early developmental human samples, and based on prior experience with similar studies in mice. No power calculations were performed to determine sample size. |
|---|---|
| Data exclusions | Data for individual cells was excluded based on quality control metrics detailed in the manuscript (Methods) |
| Replication | The reproducibility of the dataset across specimens was assessed by assessing the contribution of donors to each cluster (Supplemental Fig. 1d). The LDSC analysis linking Major Depressive Disorder to Midbrain GABAergic neurons was validated in a separate GWAS cohort. |
| Randomization | Not applicable, as we did not perform any treatment vs control experiments. |
| Blinding | The investigators were not blinded, as this was an exploratory study with anonymous untreated samples. |

# Reporting for specific materials, systems and methods

We require information from authors about some types of materials, experimental systems and methods used in many studies. Here, indicate whether each material, system or method listed is relevant to your study. If you are not sure if a list item applies to your research, read the appropriate section before selecting a response.

## Materials & experimental systems

| n/a | Involved in the study |
|---|---|
| ☒ ☐ | Antibodies |
| ☒ ☐ | Eukaryotic cell lines |
| ☒ ☐ | Palaeontology and archaeology |
| ☒ ☐ | Animals and other organisms |
| ☒ ☐ | Clinical data |
| ☒ ☐ | Dual use research of concern |
| ☒ ☐ | Plants |

## Methods

| n/a | Involved in the study |
|---|---|
| ☒ ☐ | ChIP-seq |
| ☒ ☐ | Flow cytometry |
| ☒ ☐ | MRI-based neuroimaging |

