## [Peer Review File · Nature]

Manuscript Title: Chromatin accessibility during human first-trimester neurodevelopment

Reviewer Comments & Author Rebuttals

Reviewer Reports on the Initial Version:

Referees' comments:

Referee #1 (Remarks to the Author):

Dynamics of chromatin accessibility during human first-trimester neurodevelopment

Mannens et al. investigated the accessibility of chromatin in the first trimester developing human brain using single cell ATAC seq. This study represents a monumental work of exceptionally high value to the scientific community as a resource. The major driving strength is the scope of work profiling accessibility and gene expression with a large sample size, and an exceptionally innovative utilization of the data to identify contribution of gene expression, chromatin accessibility, and individual nucleotides/TF motifs to differentiation trajectories during first trimester development.

I think that this is a wonderful resource, and would be highly appropriate for publication after a few minor revisions for clarity. Many figures and figure panels are densely packed and difficult to see/appreciate. I would be greatly appreciated if the text and graphs could be enlarged. The first section of the paper is especially difficult to follow with figures, where subpanels feel out of order with their corresponding description in the text.

Other, more specific minor comments:

Chromatin accessibility in the first trimester

- Figure 1

- o B feels like it could be in the supplement. Also what are the colors? A time trajectory with cell type composition over development illustrated before this panel would be useful instead of indicating above each PCW what stage of development it is

- o Some clusters are very biased by age of sample (c-e), how can this be separated by cell-type specific differences?

- o F is hard to interpret, even some dotted lines or colored columns to separate cell types would be helpful so we can track with TF motifs are enriched in which cell type

- ♣ The violin plots for accessibility and gene expression are also hard to see, is it possible to represent with a dot plot as well or something else?

- o Some explanation in the text also of how the enriched TF motifs are expected for the cell types they see would be helpful. And discussing how the gene expression/accessibility differs between cell types (and how expression and accessibility correlate) would be helpful too. Very little explanation of this figure overall in the text

- Extended figure 4

- o Colors in A are hard to tell apart, can't really see which is TSS. Could also be mitigated by larger graph

- o Gene track schematic in B is way too small
- o Explanation of this figure does not follow logic or flow of text (for example, jumping from explaining 4b to 4i without explaining any subpanels in between)
- ♣ Also order of extended data figures, could be worth switching extended figures 3 and 4 to follow discussion in text
- Extended figure 3
- o Gene names in F are too small
- o The difference between radial glia and glioblast and this could be worth explaining
- Text comments
- o Paragraph from lines 103-111 does not make sense to me
- ♣ Mention an increase in accessible regions and then says this is line with a shift towards heterochromatin in the next sentence (could be a difference in cell types they are commenting on but this is not clear)
- ♣ Mention an increase in accessible regions in all cell types except radial glia, and that these sites correspond with NFI-binding sites
- Then say that NFI-binding sites in radial glia more associated with mature cell neighborhoods. Do they mean that presence of these sites indicates cell types/states further along the developmental trajectory past the radial glia stage? This is not clear
- o Would be good if this section wrapped up with some statement about what these findings mean for development. Contextualizing the findings here would add a lot to the manuscript, instead of simply describing the figures

Cis-Regulatory elements predict gene expression and Transcription Factor specificity

- Figure 2
- o Again order of subpanels does not follow discussion in text and would be helpful if it did
- o Subpanel B: gene names again are too small to read
- o Would be nice if D was the same size as E for consistency
- ♣ Way it currently is it is hard to see the diagram showing mRNA transcripts in D
- o H is hard to see, specifically can't tell which topic corresponds with which motif
- o Also cannot read genes in J
- Text comments:
- o Again wrapping up this section relating the findings to development would be nice to see

Enhancer logic in neuronal specification

- Figure 3
- o Subpanel A is too small
- o Colors of cell type names are hard to see, especially tel glut
- Text/general comments
- o They mention that the GATA2 motif is present but the gene is not expression and say that it is likely DMBX1 that is important for the purkinje lineage due to motif similarity and high expression in these cells (lines 172-178). How do they know that other TFs with similar motifs, and potentially similar expression profiles, are not driving differentiation of other lineages?

Gene regulatory dynamics in Purkinje Neuron development

- Figure 4

- o Should subpanel B come after C-D?
- o E is too small to interpret
- o Maybe D should be split up with simulated expression as its own subpanel? Felt distracting when looking at the figure while reading the text, when subpanel D is referenced before the simulated expression is explained
- o F again is too small and hard to interpret
- ♣ Same with G

Chromatin accessibility and GWAS polymorphisms predict cellular targets in neuropsychiatric disorders

- Figure 5
 - o In subpanel C, how do we know that multiple cell types are not affected, since NFIA binding motif is also enriched, for example
 - ♣ Showing a dot plot of the gene expression for TFs with enriched motifs in MDD would be helpful to see here
 - ♣ Since OTX2 is one of the only TFs with expression specific to one cell type, it might be a stretch to assume this is the only cell type effected
 - o Despite this previous comment, it is nice to see in subpanel D that an MDD SNP does overlap with an OTX2 binding site in midbrain GABAergic neurons
- Text/general comments:
 - o Conclusion from this section feels like it is lacking support. I don't disagree that midbrain GABAergic neurons are implicated in MDD, but I am wondering if other cell types might be also, and if the MDD SNPs might show up in binding sites of TFs common to other cell types (were the binding sites for these TFs but unique to other cell types overlapped with MDD SNPs?)
 - o No mention of Mulvey 2023 (<https://doi.org/10.1016/j.biopsych.2023.02.009>) which implicates hippocampus and excitatory neurons in MDD. Could be difference of developmental time, but a comparison here would be nice to see

Referee #2 (Remarks to the Author):

Mannens et al. present the manuscript: "Dynamics of chromatin accessibility during human first-trimester neurodevelopment". This is an interesting manuscript that applies single cell open chromatin and multiome analysis on developing human brain tissues from between 6 post conception weeks and 13 post conception weeks. In this study, tissues were divided into major anterior to posterior developing brain areas, and the nuclei were isolated from each area to gain coverage across the entire developing brain with higher coverage of brain stem. This data set is derived from 18 donors, generated ~520,000 open chromatin profiles, and >160,000 with paired scRNA-seq and scATAC-seq allowing inference of gene expression across the dataset. They identified 135 clusters that varied across brain region and developmental age. From this dataset they conduct several types of analysis including; 1) identification of >100,000 cell-type specific open chromatin regions across developmental time, with strong overlap with the VISTA enhancers, 2) functional annotation of open chromatin regions, identification of enriched transcription factor binding sites,

and their dynamics over development, 3) identification of sequences and TF binding site that predicted several cell lineages using a CNN model, 4) description of the dynamic role of ESRRB in the Purkinje cell developmental and description of its regulatory network, and 5) association of the CREs with disease-associated SNPs to identify cell types and brain regions associated with several neuropsychiatric diseases.

This is an important study that will provide valuable data and analyses to the field. The brain-wide sampling strategy, and the use of first trimester human brain tissues make this an important dataset that will likely be heavily used by the field. They apply several analysis tools to document the CREs, describe their dynamics, and explain the regulatory logic of cell development and disease. They show how this data can be mined, and using available computational tools, to generate new predictions and reveal new biology of human neural development and neuropsychiatric diseases. Despite the strength of this study, there are several elements that could be improved.

The manuscript would be strengthened by addressing how the developing cell types correspond to mature adult brain cell types, particularly in cortex where substantial profiling has been conducted. There should also be some discussion concerning how the 135 cell types from whole human first trimester brain corresponds to the >5000 cell types from mouse whole-brain studies, or human adult brain epigenetic studies. Even focusing on diversification of one brain area like cortex would be helpful. How does this dataset help us better understand the diversification of human brain cell types?

The association of the disease associated SNPs with cell types was very interesting, but is the early developmental data critical for these insights? Could the same cell type associations be made from adult mid-brain epigenetic data? Are the key epigenetic marks dynamic or stable through adulthood? Along those lines, can the author show clear examples where identification of a disease associated cell type required using the early developmental data? Or where a key insight came from a CRE that was accessible early and vanished later in development?

Minor issues

1. The manuscript makes heavy use of many existing computational tools and they become confusing going through the manuscript. The authors call them out, but often do not spend much time describing them in simple terms, the main objectives of the tools, and the assumptions of each tool. For instance, BoolODE is only mentioned in the legend and is not explained.
2. Some of the data in figures isn't explained. For instance, in 4G, what is the "contribution score" and what does "saturation mutagenesis" mean (typo)?
3. Be clear if cells or nuclei were profiled. The writing is referencing cells, but it appears the dissociation protocol uses nuclei.
4. Extended F1G: legend has shifted on top of panel F.
5. Extended figures appear out of order in the text. They should be re-ordered to appear

sequentially.

6. Text and images on figures are often too small to read on printed versions of the figures. Please make figures and text legible.

7. Extended figure 4K: in legend is called out as G.

Author Rebuttals to Initial Comments:

We thank the reviewers for their encouraging and constructive comments. In response, we have revised the manuscript and figures, included additional supporting data, and responded to each of the specific comments below.

Referee #1 (Remarks to the Author):

Dynamics of chromatin accessibility during human first-trimester neurodevelopment
Mannens et al. investigated the accessibility of chromatin in the first trimester developing human brain using single cell ATAC seq. This study represents a monumental work of exceptionally high value to the scientific community as a resource. The major driving strength is the scope of work profiling accessibility and gene expression with a large sample size, and an exceptionally innovative utilization of the data to identify contribution of gene expression, chromatin accessibility, and individual nucleotides/TF motifs to differentiation trajectories during first trimester development.

I think that this is a wonderful resource, and would be highly appropriate for publication after a few minor revisions for clarity. Many figures and figure panels are densely packed and difficult to see/ appreciate. I would be greatly appreciated if the text and graphs could be enlarged. The first section of the paper is especially difficult to follow with figures, where subpanels feel out of order with their corresponding description in the text.

We have taken care to make the figures more readable by increasing text size and size of plots where possible. In addition, for the first section of the paper we have adjusted the order and references to figure 1 to improve the flow of the text.

Other, more specific minor comments:

• Figure 1

- **B feels like it could be in the supplement. Also what are the colors? A time trajectory with cell type composition over development illustrated before this panel would be useful instead of indicating above each PCW what stage of development it is**

We have moved panel B to extended figure 1, and added a legend for the colors (they show regions). We instead added a new plot to Figure 1 showing temporal changes in cell class composition in the sampled anatomical regions.

- **Some clusters are very biased by age of sample (c-e), how can this be separated by cell-type specific differences?**

Due to the ongoing addition of new progeny during development, the composition of cell types is constantly changing during early development. For instance, many neurons have lower average ages in this dataset than the radial glia/glioblasts. This is in part due to the switch from neurogenesis to gliogenesis that radial glia undergo at different points of time in different regions. The number of neurons does not actually decrease, but their relative frequency in comparison to glia does decrease.

One of the most “age-biased” classes were the immune cells, but this is because those cells are born outside the brain and migrate into the parenchyma during the time period we studied. Thus only at the later timepoints e.g. PCW 13 should we expect to observe microglia in the human developing brain. Furthermore, the whole brain matures faster in the posterior parts, such that at any given age, the hindbrain is more mature than the forebrain. This is now also evident in new Fig 1e.

Moreover, there are almost no stable cell types during development, most cells exist in a transitory state along their differentiation trajectory, meaning that the “same” cell type sampled at different developmental timepoints will be quite different and are likely to cluster separately. Thus we don’t think it’s possible to fully disentangle age and cell type, at least not in a 2D embedding of transcriptomes. We hope that showing all these different viewpoints in Fig 1b-e can give the reader an overall appreciation for these complexities.

- **F is hard to interpret, even some dotted lines or colored columns to separate cell types would be helpful so we can track with TF motifs are enriched in which cell type**

Dotted lines have been added at the major breaks in the dendrogram to improve the reader's orientation in the figure. We also think changing from violin plots to dot plots as suggested (next item below) made the figure much more readable.

- **The violin plots for accessibility and gene expression are also hard to see, is it possible to represent with a dot plot as well or something else?**

We have replaced the violin plots with dot-plots showing expression level and number of non-zero cells. We believe it has improved the readability as per the reviewer's suggestion. Note that the meaning of the dots differs between the sub-panels, which is indicated by the different legends for size and color in each sub-panel.

- **Some explanation in the text also of how the enriched TF motifs are expected for the cell types they see would be helpful. And discussing how the gene expression/accessibility differs between cell types (and how expression and accessibility correlate) would be helpful too. Very little explanation of this figure overall in the text.**

To address this concern, we have added two text sections discussing first the similarity between expression and accessibility of marker genes (a more detailed exploration of correlation between the two modalities can be seen in extended figure 6a-c) and a second section explaining more of the expected and observed TF enrichments displayed in fig 1f.

- **Extended figure 4**

- **Colors in A are hard to tell apart, can't really see which is TSS. Could also be mitigated by larger graph**

We agree, and have now removed the outer ring (which was anyway not very informative), allowing the inner ring to become larger and easier to read.

- **Gene track schematic in B is way too small**

Size of schematic is increased.

- **Explanation of this figure does not follow logic or flow of text (for example, jumping from explaining 4b to 4i without explaining any subpanels in between)**

We added a note in the text that indicates that 4c-f provide more details about the radial glia/glioblast shift.

- **Also order of extended data figures, could be worth switching extended figures 3 and 4 to follow discussion in text**

We have switched extended figures 3 and 4 as requested.

- **Extended figure 3**

- **Gene names in F are too small**

Font has been adjusted accordingly.

- **The difference between radial glia and glioblast and this could be worth explaining.**

We have added a sentence in the main text explaining the difference between radial glia and glioblasts.

- **Text comments**

- **Paragraph from lines 103-111 does not make sense to me**
- **Mention an increase in accessible regions and then says this is line with a shift towards heterochromatin in the next sentence (could be a difference in cell types they are commenting on but this is not clear)**
- **Mention an increase in accessible regions in all cell types except radial glia, and that these sites correspond with NFI-binding sites**
- **Then say that NFI-binding sites in radial glia more associated with mature cell neighborhoods. Do they mean that presence of these sites indicates cell types/states further along the developmental trajectory past the radial glia stage? This is not clear**
- **Would be good if this section wrapped up with some statement about what these findings mean for development. Contextualizing the findings here would add a lot to the manuscript, instead of simply describing the figures**

This section has been reworked to clarify the intended statements. In brief we observe an increase in fragments in the neuronal lineage but not in the glial lineage, which is in line with previous findings. Moreover, we identified NFI factors to be closely linked to maturation across neural cell classes.

Cis-Regulatory elements predict gene expression and Transcription Factor specificity

• Figure 2

- **Again order of subpanels does not follow discussion in text and would be helpful if it did**
Order of panels has been fixed and now follows the text.
- **Subpanel B: gene names again are too small to read**
Font size has been increased.
- **Would be nice if D was the same size as E for consistency**
- **Way it currently is it is hard to see the diagram showing mRNA transcripts in D**
D and E are now the same size, making it easier to compare the two panels
- **H is hard to see, specifically can't tell which topic corresponds with which motif**
The size of panel H (now I) has been increased allowing the topics to be shown directly next to the relevant rows of the figure.
- **Also cannot read genes in J**
Text size has also been increased (now g)

Text comments:

- **Again wrapping up this section relating the findings to development would be nice to see**
A concluding section was added to the text.

Enhancer logic in neuronal specification

• Figure 3

- **Subpanel A is too small.**
Size of panel A is increased.
- **Colors of cell type names are hard to see, especially tel glut.**
Tel glut was changed to blue. Cell type names are now in bold.

• Text/general comments

- **They mention that the GATA2 motif is present but the gene is not expression and say that it is likely DMBX1 that is important for the purkinje lineage due to motif similarity and high expression in these cells (lines 172-178). How do they know that other TFs with similar motifs, and potentially similar expression profiles, are not driving differentiation of other lineages?**
We used Jeff Vierstra's database of archemotifs to identify all motifs matching the sequences outputted by TF-MoDisco (semi-manually). This is not an exhaustive annotation, but to our knowledge the largest, best annotated reference of shared binding-motifs. This often narrows the list down to a few candidates that match the binding motif closely. We then plotted gene expression of the likely candidates and selected the correct gene name based on which TFs were indeed actively transcribed. In this case we were surprised to see the OTX2 motif in Purkinje neurons as it's a well-established midbrain marker, which was a reason for us to note the expression of DMBX1 in the text.

Gene regulatory dynamics in Purkinje Neuron development

• Figure 4

- **Should subpanel B come after C-D?**
We have kept the order of B-D as this is important for the order of events. The transcription factor network identified in 4b serves as the foundation for the simulated data in 4c-d and should be introduced and explained beforehand.
- **E is too small to interpret**
We have taken care to improve readability on panel e (now f). It's size has been increased, the indication lines have been moved to the background of the figure and their colors were softened. Finally, the signal tracks are now completely opaque.

- **Maybe D should be split up with simulated expression as its own subpanel? Felt distracting when looking at the figure while reading the text, when subpanel D is referenced before the simulated expression is explained.**
We have kept panel d together, but removed the early reference to it so that it is not mentioned before panels a-c.
- **F again is too small and hard to interpret**
Panel f (now e) has been increased in size slightly and text size has also been increased.
- **Same with G**
We have slightly decreased the size of panel g but have also cropped out large parts of the sequence that did not confer any information, leaving a smaller region that could be zoomed in on further.

Chromatin accessibility and GWAS polymorphisms predict cellular targets in neuropsychiatric disorders

• Figure 5

- **In subpanel C, how do we know that multiple cell types are not affected, since NFIA binding motif is also enriched, for example**
The identification of other TF binding sites indeed suggests that midbrain GABAergic neurons are not the only cell types involved in MDD disease pathology, which was not our intended claim. Instead we make the argument that they are selectively vulnerable during the timeframe from which we have sampled i.e. more likely to be affected by the genetic burden of MDD associated SNPs. We have taken more care to put the findings in a broader context.
- **Showing a dot plot of the gene expression for TFs with enriched motifs in MDD would be helpful to see here**
We have added a dot plot of all identified transcription factors for each cell type in supplemental figure 9b.
- **Since OTX2 is one of the only TFs with expression specific to one cell type, it might be a stretch to assume this is the only cell type effected.**
In addition to OTX2, we also identified GATA2 motifs, which are rather specific to the TAL2+ GABAergic neurons and MEIS2 motifs, which is expressed in some but not all neurons.
- **Despite this previous comment, it is nice to see in subpanel D that an MDD SNP does overlap with an OTX2 binding site in midbrain GABAergic neurons**

Text/general comments:

- **Conclusion from this section feels like it is lacking support. I don't disagree that midbrain GABAergic neurons are implicated in MDD, but I am wondering if other cell types might be also, and if the MDD SNPs might show up in binding sites of TFs common to other cell types (were the binding sites for these TFs but unique to other cell types overlapped with MDD SNPs?)**
We provide some additional analysis below where we did motif discovery on all included SNPs (847 total) after extending them with 8bp to either side. We found that they overlap a broad collection of TFs with relatively low statistical significance, many of which are broadly expressed in the brain (ZNF652, ZNF682, SOX2, FOXK1), while some of the others are rather specific (PITX3 in dopaminergic neurons, POU4F3 in midbrain glutamatergic and EOMES in cortical neuroblasts). Our conclusion from this is that indeed other cell types will be involved in disease aetiology, although we are not able to see these effects at this age in chromatin accessibility.

- No mention of Mulvey 2023 (<https://doi.org/10.1016/j.biopsych.2023.02.009>) which implicates hippocampus and excitatory neurons in MDD. Could be difference of developmental time, but a comparison here would be nice to see**
We have added a reference to Mulvey 2023 and Li 2022 (adult atlas also looking at GWAS) and discussed relevant differences in biological data and prediction results.

Referee #2 (Remarks to the Author):

Mannens et al. present the manuscript: “Dynamics of chromatin accessibility during human first-trimester neurodevelopment”. This is an interesting manuscript that applies single cell open chromatin and multiome analysis on developing human brain tissues from between 6 post conception weeks and 13 post conception weeks. In this study, tissues were divided into major anterior to posterior developing brain areas, and the nuclei were isolated from each area to gain coverage across the entire developing brain with higher coverage of brain stem. This data set is derived from 18 donors, generated ~520,000 open chromatin profiles, and >160,000 with paired scRNA-seq and scATAC-seq allowing inference of gene expression across the dataset. They identified 135 clusters that varied across brain region and developmental age. From this dataset they conduct several types of analysis including; 1) identification of >100,000 cell-type specific open chromatin regions across developmental time, with strong overlap with the VISTA enhancers, 2) functional annotation of open chromatin regions, identification of enriched transcription factor binding sites, and their dynamics over development, 3) identification of sequences and TF binding site that predicted several cell lineages using a CNN model, 4) description of the dynamic role of ESRRB in the Purkinje cell developmental and description of its regulatory network, and 5) association of the CREs with disease-associated SNPs to identify cell types and brain regions associated with several neuropsychiatric diseases.

This is an important study that will provide valuable data and analyses to the field. The brain-wide sampling strategy, and the use of first trimester human brain tissues make this an important dataset that will likely be heavily used by the field. They apply several analysis tools to document the CREs, describe their dynamics, and explain the regulatory logic of cell development and disease. They show how this data can be mined, and using available computational tools, to generate new predictions and reveal new biology of human neural development and neuropsychiatric diseases. Despite the strength of this study, there are several elements that could be improved.

The manuscript would be strengthened by addressing how the developing cell types correspond to mature adult brain cell types, particularly in cortex where substantial profiling has been conducted. There should also be some discussion concerning how the 135 cell types from whole human first trimester brain corresponds to the >5000 cell types from mouse whole-brain studies, or human adult brain epigenetic studies. Even focusing on diversification of one brain area like cortex would be helpful. How does this dataset help us better understand the diversification of human brain cell types?

Unfortunately, we don't think it's possible to meaningfully align these very early cell states with adult cell types. At the ages we sampled, for example, final cortical layers have not yet formed and we observe only immature neurons of generally deep-layer or superficial-layer type. The distance in developmental time, and thus in cell differentiation and gene expression, between first trimester embryos and fully adult humans is enormous.

Nevertheless, we now conducted a comparison between our dataset and the adult scATAC-seq dataset of the human brain from by Li et al., (<https://doi.org/10.1101/2022.11.09.515833>); the results are shown below. Here we calculated the correlation between cell types using the 10,000 most variable peaks (co-ranking) that occur in both datasets. While we can see general correlation trends between classes (neuron types to neuron types, oligodendrocytes to oligodendrocytes, astrocytes to radial glia/glioblast and immune to immune), we did not see any strong links between the neural subtypes. We expect the reason for this to be the large developmental distance between the adult state and first trimester state.

We believe (partially based on experience from mouse) that accurate matching of developmental to adult cell types will require systematic sampling of the intervening timepoints, including later gestational stages, through childhood and adolescence.

The association of the disease associated SNPs with cell types was very interesting, but is the early developmental data critical for these insights? Could the same cell type associations be made from adult mid-brain epigenetic data?

A similar approach was again taken by Li et al., (<https://doi.org/10.1101/2022.11.09.515833>) who similarly tested selective vulnerability of (adult) cell types for SNPs associated with psychiatric disease. Interestingly, they found that most psychiatric diseases they tested (including MDD) were primarily associated with cortical projecting neurons. While their data is primarily of cortical origin, it does include one sample of midbrain data and these neurons (CNMIX) were not associated with MDD. As such the temporal component does seem to be important, because not all SNPs will fall within active regulatory elements at all time points. More comprehensive and brain-wide adult human datasets would be required to fully answer the question.

Are the key epigenetic marks dynamic or stable through adulthood? Along those lines, can the author show clear examples where identification of a disease associated cell type required using the early developmental data? Or where a key insight came from a CRE that was accessible early and vanished later in development?

A key insight into the development of intellectual disability was recently given in De Vas et al., (doi: 10.26508/lsa.202201843), where they found that de novo mutations in Intermediate progenitor cell (IPC), a key cell type in neurogenesis, drive disease aetiology. Moreover, the lack of annotation for developmental enhancers compared to adult makes it difficult to identify such links and we hope the dataset presented here will help other researcher identify more of them.

In the case of our MDD associated regions, we compared them to the previously mentioned midbrain GABAergic neurons from Li et al., which is a small cluster, but it is the closest adult comparison of the same technique we could find (figure below). Most of the associated regions we identified do not seem to be active in the adult counterpart:

Minor issues

1. The manuscript makes heavy use of many existing computational tools and they become confusing going through the manuscript. The authors call them out, but often do not spend much time describing them in simple terms, the main objectives of the tools, and the assumptions of each tool. For instance, BoolODE is only mentioned in the legend and is not explained.

Where possible we have improved the description of computational tools.

2. Some of the data in figures isn't explained. For instance, in 4G, what is the "contribution score" and what does "saturation mutatgenesis" mean (typo)?

We have added an explanation of the 'contribution score' to the main text and 'saturation mutagenesis' (this was indeed a typo) to the figure legend.

3. Be clear if cells or nuclei were profiled. The writing is referencing cells, but it appears the dissociation protocol uses nuclei.

We have replaced all incorrect references to cells with nuclei.

4. Extended F1G: legend has shifted on top of panel F.

Legend has been moved back.

5. Extended figures appear out of order in the text. They should be re-ordered to appear sequentially.

We have reordered the figures in particular extended figures 3 and 4 have been switched. Extended figures are now referenced in order in the main text.

6. Text and images on figures are often too small to read on printed versions of the figures. Please make figures and text legible.

The text and size of figure elements has been increased where possible to make them more legible.

7. Extended figure 4K: in legend is called out as G.

Adjusted panel name.

8. Fig. 5: Please spell out figure acronyms in the legend.

We have spelled out the names in the figure itself.

Reviewer Reports on the First Revision:

Referees' comments:

Referee #1 (Remarks to the Author):

The authors have addressed my critiques and I do not have additional suggestions.

Referee #2 (Remarks to the Author):

Congratulation on the excellent study, Mannens et. al. The authors have amended text and figures in response to the critiques and the manuscript is in good shape. This manuscript and associated datasets will be highly valuable to the field.

The authors say and demonstrate that the developmental time between their prenatal cell types and adult cell types is too great to make meaningful connections. A brief discussion about the challenges making connections between neurons in early development and their adult counterparts, and the major chromatin changes between ages, would likely be of interest to many readers.